# SIMILAR: Submodular Information Measures Based Active Learning In Realistic Scenarios

**Suraj Kothawade**
University of Texas at Dallas
suraj.kothawade@utdallas.edu

**Nathan Beck**
University of Texas at Dallas
nathan.beck@utdallas.edu

**Krishnateja Killamsetty**
University of Texas at Dallas
krishnateja.killamsetty@utdallas.edu

**Rishabh Iyer**
University of Texas at Dallas
rishabh.iyer@utdallas.edu

## Abstract

Active learning has proven to be useful for minimizing labeling costs by selecting the most informative samples. However, existing active learning methods do not work well in realistic scenarios such as imbalance or rare classes, out-of-distribution data in the unlabeled set, and redundancy. In this work, we propose SIMILAR (**S**ubmodular **I**nformation **M**easures based act**I**ve **LeAR**ning), a unified active learning framework using recently proposed submodular information measures (SIM) as acquisition functions. We argue that SIMILAR not only works in standard active learning but also easily extends to the realistic settings considered above and acts as a *one-stop* solution for active learning that is scalable to large real-world datasets. Empirically, we show that SIMILAR significantly outperforms existing active learning algorithms by as much as $\approx 5\% - 18\%$ in the case of rare classes and $\approx 5\% - 10\%$ in the case of out-of-distribution data on several image classification tasks like CIFAR-10, MNIST, and ImageNet. SIMILAR is available as a part of the DISTIL toolkit: https://github.com/decile-team/distil.

## 1 Introduction

Deep neural networks (DNNs) have had a lot of success in a wide variety of domains. However, they require large labeled datasets which are often taxing, time-consuming, and expensive to obtain. Active learning (AL) [12, 13, 39, 3, 9] is a promising approach to solve this problem. It aims to select the most informative data points from an unlabeled dataset to be labeled in an adaptive manner with a human in the loop. The goal of AL is to achieve maximum accuracy of the model while minimizing the number of data points required to be labeled.

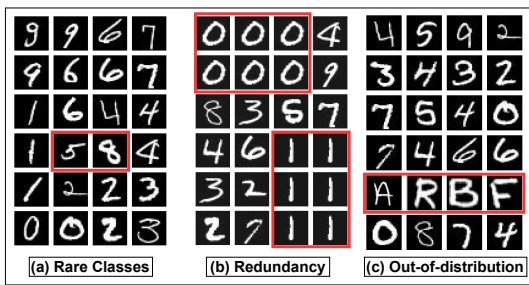

Figure 1: Motivating scenarios for realistic active learning: (a) rare classes: digits 5 and 8 are rare; (b) redundancy: digits 0 and 1 are redundant; (c) out-of-distribution (OOD): letters A, R, B, F in digit classification.

Current AL methods have been tested in relatively simple, clean, and balanced datasets. However, real-world datasets are not clean and have a number of characteristics that makes learning from them challenging [10, 46, 47, 38, 1, 8]. Firstly, these real-world datasets are *imbalanced*, and some classes are *very rare* (e.g., Fig 1(a)). Examples of this imbalance are medical

35th Conference on Neural Information Processing Systems (NeurIPS 2021).

imaging domains where the *cancerous* images are rare. Secondly, real-world data has a lot of *redundancy* (e.g., Fig 1(b)). This redundancy is more prominent in datasets that are created by sampling frames from videos (e.g., footage from a car driving on a freeway or surveillance camera footage). Thirdly, it is common to have *out-of-distribution* (OOD) (e.g., Fig 1(c)) data, where some part of the unlabeled data is not of concern to the task at hand. Given the amount of unlabeled data, it is not realistic to assume that these datasets can be cleaned manually; hence, it is the need of the hour to have active learning methods that are robust to such scenarios. We show that current AL approaches (including the state-of-the-art approach BADGE [3]) do not work well in the presence of the dataset biases described above. In this work, we address the following question: *Can a machine learning model be trained using a single unified active learning framework that works for a broad spectrum of realistic scenarios?* As a solution, we propose SIMILAR[1], a unified active learning framework which enables active learning for many realistic scenarios like rare classes, out-of-distribution (OOD) data, and redundancy.

## 1.1 Related Work

Active learning has enabled efficient training of complex deep neural networks by decreasing labeling costs. The most commonly used approach is to select the most uncertain items. Examples of uncertainty strategies include ENTROPY [41], LEAST CONFIDENCE [44], and MARGIN [37]. One challenge of this approach is that all the samples within a batch can be potentially similar even though they are uncertain. To overcome this problem in batch active learning, many recent works have attempted to select diverse yet informative data points. [45, 22] propose a simple approach: Filter a set of points using uncertainty sampling and then select a diverse subset from the filtered set. [40] propose CORESET, which forms core-sets using greedy $k$-center clustering while maintaining the geometric arrangement. BADGE [3], another recent approach, proposes to select data points corresponding to high-magnitude, diverse hypothesized gradients by using K-MEANS++ [2] initialization to distance from previously selected data points in the batch. Most existing AL approaches fail to ensure diversity across AL selection rounds and do not perform as well when there is a lot of redundancy. Sinha et al. [42] used a variational autoencoder (VAE) [25] to learn a feature space and an adversarial network [32] to distinguish between labeled and unlabeled data points. However, their approach is computationally expensive and requires extensive hyperparameter tuning. Similarly, BATCHBALD [26] does not scale to larger batch sizes since their method would need a large number of Monte Carlo dropout samples to obtain a significant mutual information. Such limitations reduce the scope of applying these methods to realistic settings.

Closely related to our work are two recently proposed works. The first is GLISTER-ACTIVE [24], which formulates the AL acquisition function by maximizing the log-likelihood on a held-out validation set. This validation set could consist of examples from the rare classes or in-distribution examples. The second approach is the work of Gudovskiy et al. [15], who study AL for biased datasets using a self-supervised FISHER kernel and pseudo-label estimators. They address this problem by explicitly minimizing the KL divergence between training and validation sets via maximizing the FISHER kernel. Although their method shows promising results, they make multiple unrealistic assumptions: a) They use a *large labeled validation set*, and b) they use feature representations from a model pretrained using unsupervised learning on a *balanced* unlabeled dataset. In this work, we compare against both GLISTER-ACTIVE [24] and FISHER [15] approaches in the more realistic setting of a small held-out validation set (smaller than the seed labeled set) and an imbalanced unlabeled set. Another work proposed a discrete optimization method for $k$-NN-type algorithms in the domain shift setting [6]. However, their approach is limited to $k$-NNs.

This work utilizes submodular information measures (SIM) by [19] and their extensions by [23]. SIMs encompass submodular conditional mutual information (SCMI), which can then be used to derive submodular mutual information (SMI); submodular conditional gain (SCG); and submodular functions (SF). We discuss these functions in detail in Sec. 2. [23] also studies these functions on the closely related problem of targeted data selection.

## 1.2 Our Contributions

The following are our main contributions: **1)** Given the limitations of existing approaches in handling active learning in the real world, we propose SIMILAR (Sec. 3), a unified active learning framework that can serve as a comprehensive solution to multiple realistic scenarios. **2)** We treat SIM as a

---

[1]**S**ubmodular **I**nformation **M**easures based act**I**ve **L**e**AR**ning

common umbrella for realistic active learning and study the effect of different function instantiations offered under SIM for various realistic scenarios. **3)** SIMILAR not only handles standard active learning but also extends to a wide range of settings which appear in the real world such as rare classes, out-of-distribution (OOD) data, and datasets with a lot of redundancy. Finally, **4)** we empirically demonstrate the effectiveness of SMI-based measures for image classification (Sec. 4) in a number of realistic data settings including imbalanced, out-of-distribution, and redundant data. Specifically, in the case of imbalanced and OOD data, we show that SIMILAR achieves improvements of more than 5 to 10% on several image classification datasets.

## 2 Background

In this section, we enumerate the different submodular functions that are covered under SIM and the relationships between them.

**Submodular Functions.** We let $\mathcal{U}$ denote the *unlabeled* set of $n$ data points $\mathcal{U} = \{1, 2, 3, ..., n\}$ and a set function $f : 2^{\mathcal{U}} \rightarrow \mathbb{R}$. Formally, a function $f$ is submodular [14] if for $x \in \mathcal{U}$, $f(\mathcal{A} \cup x) - f(\mathcal{A}) \geq f(\mathcal{B} \cup x) - f(\mathcal{B})$, $\forall \mathcal{A} \subseteq \mathcal{B} \subseteq \mathcal{U}$ and $x \notin \mathcal{B}$. For a set $\mathcal{A} \subseteq \mathcal{U}$, $f(\mathcal{A})$ provides a real-valued score for $\mathcal{A}$. In the context of batch active learning, this is the score of an acquisition function $f$ on batch $\mathcal{A}$. Submodularity is particularly appealing because it naturally occurs in real world applications [43, 4, 5, 20] and also admits a constant factor $1 - \frac{1}{e}$ [34] for cardinality constraint maximization. Additionally, variants of the greedy algorithm maximize a submodular function in *near-linear time* [33].

**Submodular Mutual Information (SMI).** Given sets $\mathcal{A}, \mathcal{Q} \subseteq \mathcal{U}$, the SMI [16, 19] is defined as $I_f(\mathcal{A}; \mathcal{Q}) = f(\mathcal{A}) + f(\mathcal{Q}) - f(\mathcal{A} \cup \mathcal{Q})$. Intuitively, *SMI models the similarity between $\mathcal{Q}$ and $\mathcal{A}$*, and maximizing SMI will select points *similar* to $\mathcal{Q}$ while being diverse. $\mathcal{Q}$ here is the query set.

**Submodular Conditional Gain (SCG).** Given sets $\mathcal{A}, \mathcal{P} \subseteq \mathcal{U}$, the SCG $f(\mathcal{A}|\mathcal{P})$ is the gain in function value by adding $\mathcal{A}$ to $\mathcal{P}$. Thus, $f(\mathcal{A}|\mathcal{P}) = f(\mathcal{A} \cup \mathcal{P}) - f(\mathcal{P})$ [19]. Intuitively, SCG models how different $\mathcal{A}$ is from $\mathcal{P}$, and maximizing SCG functions will select data points *not similar to the points in $\mathcal{P}$* while being diverse. We refer to $\mathcal{P}$ as the conditioning set.

**Submodular Conditional Mutual Information (SCMI).** Given sets $\mathcal{A}, \mathcal{Q}, \mathcal{P} \subseteq \mathcal{U}$, the SCMI is defined as $I_f(\mathcal{A}; \mathcal{Q}|\mathcal{P}) = f(\mathcal{A} \cup \mathcal{P}) + f(\mathcal{Q} \cup \mathcal{P}) - f(\mathcal{A} \cup \mathcal{Q} \cup \mathcal{P}) - f(\mathcal{P})$. Intuitively, SCMI *jointly models the similarity between $\mathcal{A}$ and $\mathcal{Q}$ and their dissimilarity with $\mathcal{P}$*.

**Relationship between SIM** The relationship between the above measures is the key component that unifies our AL framework [19, 23]. The unification comes from the rich modeling capacity of SCMI: $I_f(\mathcal{A}; \mathcal{Q}|\mathcal{P})$ where $\mathcal{Q}, \mathcal{P} \subseteq \mathcal{U}$. This facilitates a *single* acquisition function that can be applied to *multiple* scenarios. Concretely, the submodular function $f$ can be obtained

| Function | Setting | Realistic Scenario |
|----------|---------|--------------------|
| Submodular | $\mathcal{Q} \leftarrow \mathcal{U}, \mathcal{P} \leftarrow \emptyset$ | Standard AL |
| SMI | $\mathcal{Q} \leftarrow \mathcal{Q}, \mathcal{P} \leftarrow \emptyset$ | Imbalance, OOD |
| SCG | $\mathcal{Q} \leftarrow \emptyset, \mathcal{P} \leftarrow \mathcal{P}$ | Redundancy |
| SCMI | $\mathcal{Q} \leftarrow \mathcal{Q}, \mathcal{P} \leftarrow \mathcal{P}$ | OOD |

Table 1: Relationship between SIM and their applications to realistic scenarios by choices of $\mathcal{Q}$ and $\mathcal{P}$.

by setting $\mathcal{Q} \leftarrow \mathcal{U}$ and $\mathcal{P} \leftarrow \emptyset$. Next, the SMI can be obtained by setting $\mathcal{Q} \leftarrow \mathcal{Q}$ and $\mathcal{P} \leftarrow \emptyset$, while we obtain SCG by setting $\mathcal{Q} \leftarrow \emptyset, \mathcal{P} \leftarrow \mathcal{P}$. We summarize the relationships between SIM in Tab. 1.

**Instantiations of SIM.** The formulations for Facility Location (FL), Graph Cut (GC) and Log Determinant (LOGDET) are as in [19, 23] and we adapt them as acquisition functions for batch active learning. We use two variants for FL: FLQMI, which models pairwise similarities of *only the query set* $\mathcal{Q}$ to the unlabeled dataset, and FLVMI, which additionally considers the pairwise similarities within the unlabeled dataset $\mathcal{U}$. The SCG and SCMI expressions corresponding to FL are referred as FLCG and FLCMI, respectively (see row 1 in Tab. 2a and 2b). For LOGDET, we refer to the SMI, SCG and SCMI expressions as LOGDETMI, LOGDETCG and LOGDETCMI, respectively (see row 5 in Tab. 2a and row 2 in Tab. 2b). Similarly, the SMI and SCG expressions are respectively referred to as GCMI and GCCG for GC (see row 3 in Tab. 2a and 2b). For notation in Tab. 2, the pairwise similarity matrix $S$ between items in sets $\mathcal{A}$ and $\mathcal{B}$ is denoted as $S_{\mathcal{A},\mathcal{B}}$. Also, we denote $S_{ij}$ as the $(i, j)$ entry of $S$.

## 3 SIMILAR: Our Unified Active Learning Framework

In this section, we propose a unified active learning framework SIMILAR, which uses SIMs to address the limitations of the current work (see Sec. 1.1). We show that SIMILAR can be effectively applied to a broad range of realistic scenarios and thus acts as *one-stop* solution for AL.

Table 2: Instantiations of SIM. Note how the relationships in Tab. 1 can be applied to SCMI instantiations to obtain SMI and SCG instantiations.

(a) Instantiations of SMI functions.

| SMI | $I_f(\mathcal{A}; \mathcal{Q})$ |
|---|---|
| FLVMI | $\sum_{i \in \mathcal{U}} \min(\max_{j \in \mathcal{A}} S_{ij}, \max_{j \in \mathcal{Q}} S_{ij})$ |
| FLQMI | $\sum_{i \in \mathcal{Q}} \max_{j \in \mathcal{A}} S_{ij} + \sum_{i \in \mathcal{A}} \max_{j \in \mathcal{Q}} S_{ij}$ |
| GCMI | $2 \sum_{i \in \mathcal{A}} \sum_{j \in \mathcal{Q}} S_{ij}$ |
| LOGDETMI | $\log \det(S_{\mathcal{A}}) - \log \det(S_{\mathcal{A}} - S_{\mathcal{A},\mathcal{Q}} S_{\mathcal{Q}}^{-1} S_{\mathcal{A},\mathcal{Q}}^T)$ |

(b) Instantiations of SCG and SCMI functions.

| SCG | $f(\mathcal{A}|\mathcal{P})$ |
|---|---|
| FLCG | $\sum_{i \in \mathcal{U}} \max(\max_{j \in \mathcal{A}} S_{ij} - \max_{j \in \mathcal{P}} S_{ij}, 0)$ |
| LogDetCG | $\log \det(S_{\mathcal{A}} - S_{\mathcal{A},\mathcal{P}} S_{\mathcal{P}}^{-1} S_{\mathcal{A},\mathcal{P}}^T)$ |
| GCCG | $f(\mathcal{A}) - 2 \sum_{i \in \mathcal{A}, j \in \mathcal{P}} S_{ij}$ |

| SCMI | $I_f(\mathcal{A}; \mathcal{Q}|\mathcal{P})$ |
|---|---|
| FLCMI | $\sum_{i \in \mathcal{U}} \max(\min(\max_{j \in \mathcal{A}} S_{ij}, \max_{j \in \mathcal{Q}} S_{ij}) - \max_{j \in \mathcal{P}} S_{ij}, 0)$ |
| LogDetCMI | $\log \frac{\det(I - S_{\mathcal{P}}^{-1} S_{\mathcal{P},\mathcal{Q}} S_{\mathcal{Q}}^{-1} S_{\mathcal{P},\mathcal{Q}}^T)}{\det(I - S_{\mathcal{A}\cup\mathcal{P}}^{-1} S_{\mathcal{A}\cup\mathcal{P},\mathcal{Q}} S_{\mathcal{Q}}^{-1} S_{\mathcal{A}\cup\mathcal{P},\mathcal{Q}}^T)}$ |

The basic idea behind our framework is to exploit the relationship between the SIMs (Tab. 1) such that it can be applied to any real-world dataset. Particularly, we use the formulation of SCMI and appropriately choose a query set $\mathcal{Q}$ and/or a conditioning set $\mathcal{P}$ depending on the scenario at hand. Towards this end, we use the inspiration from [3] where they select data points based on diverse gradients. The SIM functions (see Tab. 2) are instantiated using similarity kernels computed using pairwise similarities $S_{ij}$ between the gradients of the current model. Specifically, we define $S_{ij} = \langle \nabla_\theta \mathcal{H}_i(\theta), \nabla_\theta \mathcal{H}_j(\theta) \rangle$, where $\mathcal{H}_i(\theta) = \mathcal{H}(x_i, y_i, \theta)$ is the loss on the $i$th data point. Similar to [45, 3], we use hypothesized labels for computing the gradients, and the corresponding similarity kernels. The hypothesized label for each data point is assigned as the class with the maximum probability. We then optimize a SCMI function:

$$\max_{\mathcal{A} \subseteq \mathcal{U}, |\mathcal{A}| \leq B} I_f(\mathcal{A}; \mathcal{Q}|\mathcal{P}) \tag{1}$$

with appropriate choices of query set $\mathcal{Q}$ and conditioning set $\mathcal{P}$. In the context of batch active learning, $\mathcal{A}$ is the batch and $B$ is the budget (batch size in AL). We present our unified AL framework in Algorithm 1 and illustrate the choices of query and conditioning set for realistic scenarios in Fig. 2.

---

**Algorithm 1** SIMILAR: Unified AL Framework

---

**Require:** Initial Labeled set of data points: $\mathcal{L}$, large unlabeled dataset: $\mathcal{U}$, Loss function $\mathcal{H}$ for learning model $\mathcal{M}$, batch size: $B$, number of selection rounds: $N$
1: **for** selection round $i = 1 : N$ **do**
2:     Train model $\mathcal{M}$ with loss $\mathcal{H}$ on the current labeled set $\mathcal{L}$ and obtain parameters $\theta$
3:     Using model parameters $\theta_i$, compute gradients using hypothesized labels $\{\nabla_\theta \mathcal{H}(x_j, \hat{y_j}, \theta), \forall j \in \mathcal{U}\}$ and obtain a similarity matrix $X$.
4:     Instantiate a submodular function $f$ based on $X$.
5:     $\mathcal{A}_i \leftarrow \text{argmax}_{\mathcal{A} \subseteq \mathcal{U}, |\mathcal{A}| \leq B} I_f(\mathcal{A}; \mathcal{Q}|\mathcal{P})$ (Optimize SCMI with an appropriate choice of $\mathcal{Q}$ and $\mathcal{P}$, see Tab. 1)
6:     Get labels $L(\mathcal{A}_i)$ for batch $\mathcal{A}_i$ and $\mathcal{L} \leftarrow \mathcal{L} \cup L(\mathcal{A}_i), \mathcal{U} \leftarrow \mathcal{U} - \mathcal{A}_i$
7: **end for**
8: Return trained model $\mathcal{M}$ and parameters $\theta$.

---

In the scenarios below, we will discuss how this paradigm can provide a unified view of active learning, handle aspects like standard active learning (Sec. 3.1), rare classes and imbalance (Sec. 3.2), redundancy (Sec. 3.3) and, OOD/outliers in the unlabeled data (Sec. 3.4).

## 3.1 Standard Active Learning

We refer to standard active learning for ideal scenarios when there is no imbalance, redundancy or OOD data in the unlabeled dataset. In such cases, there is no requirement for having a query set and conditioning set. Hence, given a SCMI function $I_f(\mathcal{A}; \mathcal{Q}|\mathcal{P})$, we get $I_f(\mathcal{A}; \mathcal{Q}|\mathcal{P}) = f(\mathcal{A})$ by setting $\mathcal{Q} \leftarrow \mathcal{U}$ (the unlabeled dataset) and $\mathcal{P} \leftarrow \emptyset$. In a nutshell, the standard diversified active learning setting can be seen as a special case of our proposed unified AL framework (Equ. (1)) by choosing $\mathcal{Q}, \mathcal{P}$ as above. Note that this approach is very similar and closely related to BADGE [3], where the authors

also choose points based on diverse gradients. Furthermore, the authors discuss the use of Determinantal Point Processes (DPP) [28] for sampling, and this is very similar to maximizing log-determinants. In the supplementary paper, we compare the choice of different submodular functions for AL.

## 3.2 Rare Classes

A very common and naturally occurring scenario is that of imbalanced data. This imbalance is because some classes or attributes are naturally more frequently occurring than others in the real-world. For example, in a self-driving car application, there may be very few images of pedestrians at night on highways, or cyclists at night. Another example is medical imaging, where there are many rare yet important diseases (e.g., various forms of cancers), and it is often the case that non-cancerous images are much more than compared to the cancerous ones. While such classes are rare, it is also critical to be able to perform well in these classes. The problem with running standard active learning algorithms in such a case is that they may not sample too many data points from these rare classes, and as a result, the model continues to perform poorly on these classes. In such cases, we can create a (small) held-out set $\mathcal{R}$ which contains data points from these rare classes, and try to encourage the AL by sampling more of these rare classes by maximizing the SMI function $I_f(\mathcal{A}; \mathcal{R})$:

$$\max_{\mathcal{A} \subseteq \mathcal{U}, |\mathcal{A}| \leq B} I_f(\mathcal{A}; \mathcal{R}) \qquad (2)$$

This setting is shown in Fig. 2(a). $\mathcal{R}$ contains a small number of held-out examples of classes $5, 8$ which are rare, and the AL acquisition function is Equ. (2). Note that this is exactly equivalent to maximizing the SCMI function with $\mathcal{Q} \leftarrow \mathcal{R}$ and $\mathcal{P} \leftarrow \emptyset$ (i.e. Equ. (1) in Line 5 of Algorithm 1). Furthermore, since the SMI functions naturally model query relevance and diversity, they will also try to pick a diverse set of data points which are relevant to $\mathcal{R}$. Finally, we also point out that this setting was considered in [15] where they use a FISHER kernel based approach to sample data points. Note that for this setting to be realistic, it is critical that the size of this validation set is very small – [15] uses a much larger validation set which is not very realistic (e.g., $200\times$ our set, see Appendix B for more details).

## 3.3 Redundancy in Unlabeled Data

Another commonplace scenario is where we are dealing with a lot of redundancy – *e.g.,* frames sampled from a video, where subsequent frames are visually similar. In such cases, existing AL algorithms tend to pick data points that are semantically similar to the ones selected in some earlier batch. This is true even for the state-of-the-art AL algorithm BADGE [3] that attempts to enforce diversity, but only in the current batch of data points and not the already selected labeled set. To illustrate this, consider the scenario in Fig. 2(b). The digits $0, 1$ are redundant in the unlabeled set, and they are already present in the labeled set $\mathcal{L}$. Algorithms which just focus on diversity in the current batch could fail at ensuring diversity across batches. To mitigate inter-batch redundancy, we use SCG acquisition function and condition upon the already labeled set $\mathcal{L}$:

$$\max_{\mathcal{A} \subseteq \mathcal{U}, |\mathcal{A}| \leq B} f(\mathcal{A}|\mathcal{L}) \qquad (3)$$

Notice that this is a special case of our proposed unified AL framework (Equ. (1)) since the SCG function $f(\mathcal{A}|\mathcal{L})$ is basically a SCMI function with $\mathcal{Q} \leftarrow \emptyset$ and $\mathcal{P} \leftarrow \mathcal{L}$.

## 3.4 Out of Distribution Data

In real world scenarios, we often have out-of-distribution (OOD) data or irrelevant classes in the unlabeled set. Such OOD data is not useful for

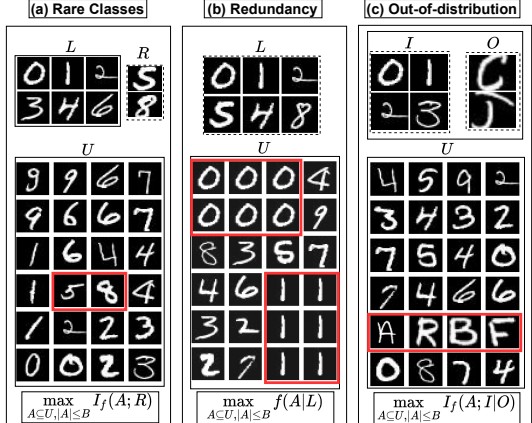

Figure 2: An illustration of realistic scenarios where SIMILAR is applied with appropriate choices of query and conditioning sets: a) SIMILAR finds rare digits $5, 8 \in \mathcal{U}$, by optimizing the SMI function $I_f(\mathcal{A}; \mathcal{R})$ with $\mathcal{R}$ containing $5, 8$ as *queries*, b) select samples from $\mathcal{U}$ which are diverse among themselves and also diverse w.r.t those in $\mathcal{L}$ by optimizing $f(\mathcal{A}|\mathcal{L})$ (here, we want to *avoid* digits $0, 1 \in \mathcal{U}$ altogether because they are present in $\mathcal{L}$), c) select digits (in-distribution) and avoid alphabets (out-of-distribution) in $\mathcal{U}$ by optimizing $I_f(\mathcal{A}; \mathcal{I}|\mathcal{O})$, where $\mathcal{I}$ are ID labeled points and $\mathcal{O}$ are OOD points selected so far.

the given classification task at hand. Using an acquisition function that selects a lot of OOD data points will lead to a waste of labeling effort and time. This is because annotators have to spend time in filtering out OOD data points and discard them from the training dataset. To account for OOD data, we add an additional class called "OOD" in our model. Since the goal is to improve on in-distribution classes , we ignore the prediction for the OOD class at test time. For our AL acquisition function, we use the currently labeled OOD points $\mathcal{O}$ as the conditioning set $\mathcal{P}$, and the currently labeled in-distribution (ID) points $\mathcal{I}$ as the query set $\mathcal{Q}$. In other words, our acquisition function is to optimize:

$$\max_{\mathcal{A} \subseteq \mathcal{U}, |\mathcal{A}| \leq B} I_f(\mathcal{A}; \mathcal{I} | \mathcal{O}) \tag{4}$$

This is illustrated in Fig. 2(c), where the labeled set consists of six examples, four of them being ID data points (set $\mathcal{I}$) and two being OOD data points (set $\mathcal{O}$). In Fig. 2(c), the ID data are digits (digit classification) and the OOD examples are alphabets. This SCMI based approach will naturally pick points "close" to the ID data while avoiding the OOD points.

Another approach for designing the acquisition function is to not explicitly condition on the OOD data points. In other words, we can just optimize the SMI function:

$$\max_{\mathcal{A} \subseteq \mathcal{U}, |\mathcal{A}| \leq B} I_f(\mathcal{A}; \mathcal{I}) \tag{5}$$

We contrast the choices of SCMI (Equ. (4)) and SMI (Equ. (5)) functions in our experiments.

### 3.5 Multiple Co-occurring Realistic Scenarios

We can also apply SIMILAR to datasets where more than one realistic scenarios are co-occurring. As illustrated in Tab. 3, we can use the formulation of SCMI and make appropriate choices of $\mathcal{Q}$ and $\mathcal{P}$ to tackle multiple realistic scenarios.

| Function | Setting | Realistic Scenario |
|---|---|---|
| $I_f(\mathcal{A}; \mathcal{R} | \mathcal{O})$ | $\mathcal{Q} \leftarrow \mathcal{R}, \mathcal{P} \leftarrow \mathcal{O}$ | Rare classes + OOD |
| $I_f(\mathcal{A}; \mathcal{R} | \mathcal{L} - \tilde{\mathcal{R}})$ | $\mathcal{Q} \leftarrow \mathcal{R}, \mathcal{P} \leftarrow \mathcal{L} - \tilde{\mathcal{R}}$ | Rare classes + Redundancy |
| $I_f(\mathcal{A}; \mathcal{I} | \mathcal{O} \cup \mathcal{I}^{'})$ | $\mathcal{Q} \leftarrow \mathcal{I}, \mathcal{P} \leftarrow \mathcal{O} \cup \mathcal{I}^{'}$ | Redundancy + OOD |

Table 3: Choices for $\mathcal{Q}$ and $\mathcal{P}$ for multiple co-occuring realistic scenarios

**Rare classes and OOD:** We set $\mathcal{Q} \leftarrow \mathcal{R}$ and $\mathcal{P} \leftarrow \mathcal{O}$ and maximize $I_f(\mathcal{A}; \mathcal{R} | \mathcal{O})$. Intuitively, this function would pick points close to $\mathcal{R}$ while avoiding the OOD points. In this scenario, we can also optimize an SMI function $I_f(A; R)$ if the data points belonging to the rare classes are not similar to the OOD data points, meaning that only searching for rare classes may suffice. Regardless, the SCMI approach above will further reinforce the avoidance of the OOD points.

**Rare classes and Redundancy:** We set $\mathcal{Q} \leftarrow \mathcal{R}$ and $\mathcal{P} \leftarrow \mathcal{L} - \tilde{\mathcal{R}}$. Here, $\tilde{\mathcal{R}}$ is the subset of data points from the labeled set $\mathcal{L}$ that belong to the rare classes. Intuitively, this function would pick points close to $\mathcal{R}$ while avoiding points already in $\mathcal{L} - \tilde{\mathcal{R}}$, thereby avoiding redundant data. Just focusing on $\mathcal{R}$ by optimizing $I_f(\mathcal{A}; \mathcal{R})$ is also a feasible option because rare classes are generally not redundant. As before, the SCMI approach will only reinforce the avoidance of redundant samples in any non-rare class instances selected.

**Redundancy and OOD:** This is a more challenging scenario than the ones above. We start with using the SCMI formulation for the OOD scenario, i.e., $I_f(\mathcal{A}; \mathcal{I} | \mathcal{O})$, where $\mathcal{I}$ is the set of ID samples and $\mathcal{O}$ is the set of OOD samples. Optimizing this function will pick diverse in-distribution samples within a batch. For selecting diverse samples across different batches, we can tackle this by using an appropriate kernel for the conditioning set. For instance, consider the FLCMI function in Tab. 2(b). On setting $\mathcal{P} \leftarrow \mathcal{O} \cup \mathcal{I}$, we can rewrite the FLCMI function by splitting the penalty term as follows: $\sum_{i \in \mathcal{U}} \max(\min(\max_{j \in \mathcal{A}} S_{ij}, \max_{j \in \mathcal{I}} S_{ij}) - \max(\max_{j \in \mathcal{O}} S_{ij}, \max_{j \in \mathcal{I}} S'_{ij}), 0)$. While $S$ is computed using cosine similarity, we can compute $S'$ using an exponential kernel to magnify the value of $S'_{ij}$ using the exponent when $i$ and $j$ are very similar. This exponent is a hyperparameter which can be tuned to penalize selecting redundant samples from $I$ (denoted as $I^{'}$) in Tab. 3.

### 3.6 Realizing Realistic Scenarios in Applications

In this section, we discuss a few insights on how these realistic scenarios can be realized. To begin with, the initial labeled set used in AL usually follows the distribution of the unlabeled set. The

statistics of this set can be used to identify rare classes. If the initial seed set is small, the rare classes/OOD data points can be realized after a few rounds of standard AL. Until such scenarios are discovered, standard AL can be done using a diversity-based acquisition function like the log determinant (LOGDET). For production-level models, they go through a test deployment phase. During this phase, systematically recurring errors are often found. An example is of undetected bicycles at night in an object detector (false negatives). Such recurring failure cases can be due to rare classes in the labeled set. Moreover, we as users often know whether there are rare classes or if there is redundancy from domain knowledge. For instance, in the biomedical domain, images of cancer cells are typically rarer than ones of non-cancer cells because cancer inherently is a rare disease.

### 3.7  Scalability and Computational Aspects of SIMILAR

**Computational Complexity:**  The computational complexity of the different SMI functions are determined by (1) the kernel computation time, and (2) the time complexity of the greedy algorithm. All functions considered here are graph based functions and require computing a kernel matrix. The LOGDET functions (LOGDET, LOGDETMI, LOGDETCG, LOGDETCMI), some FL functions (FL, FLVMI, FLCMI), and GC, GCMI all require the $n \times n$ similarity matrix ($n = |\mathcal{U}|$ is the number of unlabeled points) which entails a complexity of $O(n^2)$ to construct the similarity kernel. Once constructed, the complexity of the greedy algorithm for LOGDET class of functions is roughly $O(B^3 n)$ [11], while the complexity of the greedy algorithm with FL, FLVMI, and FLCMI is $O(Bn^2)$ [18, 20]($B$ is the batch size). Different from others, FLQMI does not require computing a $n \times n$ kernel, but only a $n \times q$ kernel (where $q = |\mathcal{Q}|$ is the number of query points). Correspondingly, the complexity of the greedy algorithm with FLQMI is $O(nqB)$, and is linear in $n$. In Appendix. A, we provide a detailed summary of the complexity of different SF, SMI, SCG, and SCMI functions.

**Partition Trick:**  The deal with the high $O(n^2)$ of the LOGDET, GC, and some of the FL variants (except FLQMI), we also propose the following partitioning algorithm: We randomly split the unlabeled set $\mathcal{U}$ into $p$ partitions $\mathcal{U}_1, \cdots, \mathcal{U}_p$, and we then define the corresponding function (SF, SMI, SCMI, SCG) on each of the partitions and independently optimize them. In each partition, we select $B/p$ points. The complexity of this reduces from $O(n^2)$ to $O(n^2/p)$ and with an appropriate choice of $p$, we can significantly reduce the computational complexity. We use this in our ImageNet experiments (see Sec. 4.1), and observe that our approaches continue performing well while being more scalable. We provide more details on partitioning in Appendix. A.

**Last Layer Gradients:**  Deep models have numerous parameters leading to very high dimensional gradients. Since our kernel matrix is computed using the cosine similarity of gradients, this becomes intractable for most models. To solve this problem, we use last-layer gradient approximation by representing data points using last layer gradients. BADGE [3], CORESET [40] and GLISTER [24] are other baselines that also use this approximation. Using this representation, we compute a pairwise cosine similarity matrix to instantiate acquisition functions in SIMILAR (see lines 3,4 in Algorithm 1).

## 4  Experimental Results

In this section, we empirically evaluate the effectiveness of SIMILAR on a wide range of scenarios like rare classes (Sec. 4.1), redundancy (Sec. 4.2) and out-of-distribution (Sec. 4.3). We do so by comparing the accuracy and selections of various SCMI based acquisition functions with existing AL approaches. Using these experiments, we cover the issues with the current AL methods and show that these issues can be mitigated by using a unified implementation using SCMI with appropriate choices of query and/or conditioning sets. Although this section focuses on realistic scenarios, we also study SIMILAR in a standard active learning setting and show that it performs at par with current AL methods (see Appendix. C). Furthermore, we present some experiments on a real-world medical dataset in Appendix. H and some experiments on multiple co-occurring realistic scenarios (Sec. 3.5) in Appendix. I.

**Baselines in all scenarios:** We compare SCMI based functions against several methods. Particularly, we compare against: (1) three uncertainty based AL algorithms: i)ENTROPY: Selects the top $B$ data points with the highest *entropy* [41], ii) MARGIN: Select the bottom $B$ data points that have the least difference in the confidence of first and the second most probable labels [37], iii)LEAST-CONF: Select $B$ samples with the smallest predicted class probability [44], (2) state-of-the-art diversity based algorithms: iv) BADGE [3] v) GLISTER [24] vi) CORESET [40] which are all discussed in section Sec. 1.1, and, 3) RANDOM: Select $B$ samples randomly. Additionally, in the rare classes scenario, we compare against FISHER [15] which is also discussed in Sec. 1.1.

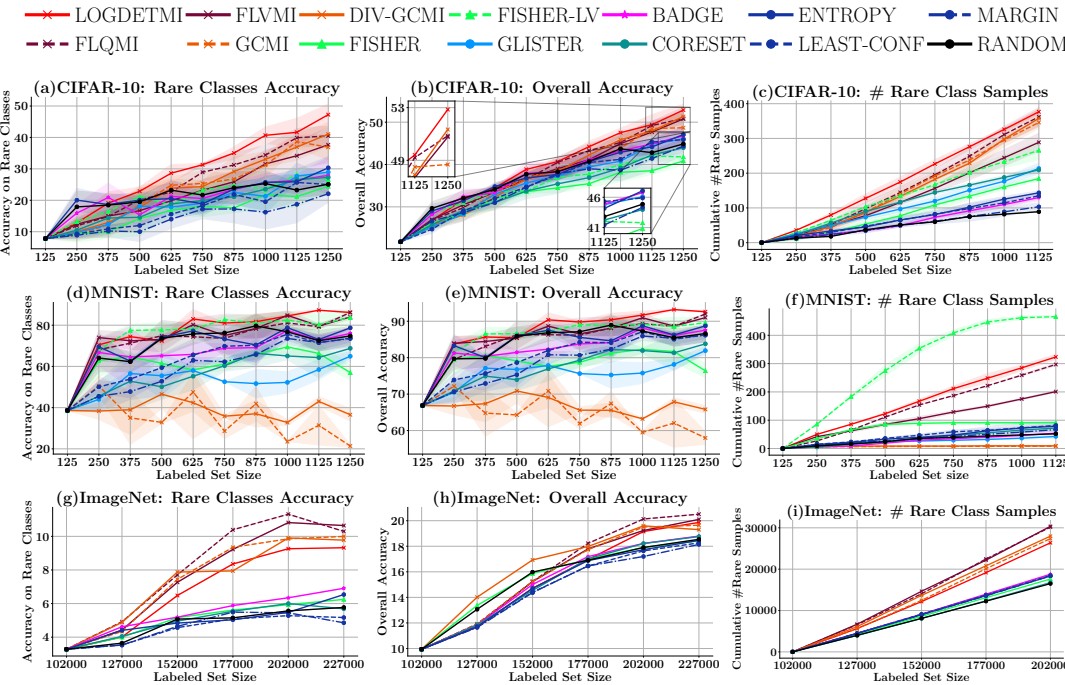

Figure 3: Active Learning with rare classes on CIFAR-10 (top row), MNIST (middle row), and ImageNet (bottom row). Left side plots (a,d,g) are rare class accuracies, center plots (b,e,h) are overall test accuracies, right plots (c,f,i) are a number of rare class samples selected. The SMI functions (specifically LOGDETMI, FLQMI) outperform other baselines by more than 10% on the rare classes.

**Datasets, model architecture and experimental setup:** We apply our framework to CIFAR-10 [27] and MNIST [30] classification tasks. Additionally, we also evaluate our method on down sampled $32\times$ 32 ImageNet-2012 [38] for the rare classes setting (Sec. 4.1). Due to the lack of test split on ImageNet, we used the validation split for evaluation. In the sections below, we discuss the individual splits for $\mathcal{L}, \mathcal{U}, \mathcal{R}, \mathcal{I}$, and $\mathcal{O}$ in each realistic scenario. To ensure that all the selection algorithms that we are studying are given fair and equal treatment across all realistic scenarios, we use a common training procedure and hyperparameters. We use standard augmentation techniques like random crop, horizontal flip followed by data normalization except for MNIST which does not use horizontal flip to preserve labels. For training, we use an SGD optimizer with an initial learning rate of 0.01, the momentum of 0.9, and a weight decay of 5e-4. We decay the learning rate using cosine annealing [31] for each epoch. On all datasets except MNIST, we train a ResNet18 [17] model, while on MNIST we train a LeNet [29] model. For all the experiments in a particular scenario (rare classes, redundancy and OOD), we start with an identical initial model $\mathcal{M}$ and initial labeled set $\mathcal{D}$. We reinitialize the model parameters at the beginning of every selection round using Xavier initialization and train the model until either the training accuracy reaches 99% or the epoch count reaches 150. We run each experiment $3\times$ on CIFAR-10 and MNIST and $1\times$ on ImageNet and provide error bars (std deviation). All experiments were run on a V100 GPU. For more details on the experimental setup, baselines, and datasets see Appendix. B.

## 4.1 Rare Classes

**Custom dataset:** Following [15, 24], we simulate these rare classes by creating a class imbalance. We initialize the batch active learning experiments by creating a custom dataset which is a subset of the full dataset with the same marginal distribution. Given that $\mathcal{C}$ consists of data points from the imbalanced classes and $\mathcal{D}$ consists of data points from the balanced classes, we create an initial labeled set $\mathcal{L}$ such that $|\mathcal{D}_{\mathcal{L}}| = \rho|\mathcal{C}_{\mathcal{L}}|$ and an unlabeled set $|\mathcal{D}_{\mathcal{U}}| = \rho|\mathcal{C}_{\mathcal{U}}|$, where $\rho$ is the imbalance factor. We use a small and clean validation/query set $\mathcal{R}$ containing data points from the imbalanced classes ($\approx 3$ data points per imbalanced class). We create an imbalance in CIFAR-10 using 5 random classes, $\rho = 10$ and for MNIST we create an imbalance using the same classes as in [15] $(5 \cdots 9)$ and use $\rho = 20$. For both datasets: $|\mathcal{C}_{\mathcal{L}}| + |\mathcal{D}_{\mathcal{L}}| = 125, |\mathcal{C}_{\mathcal{U}}| + |\mathcal{D}_{\mathcal{U}}| = 16.5K, B = 125$ (AL batch size) and, $|\mathcal{R}| = 25$ (size of the held out rare instances). For MNIST, we also present the results for $B = 25$ and $\rho = 100$ in the supplementary. On ImageNet, we randomly select 500 classes out

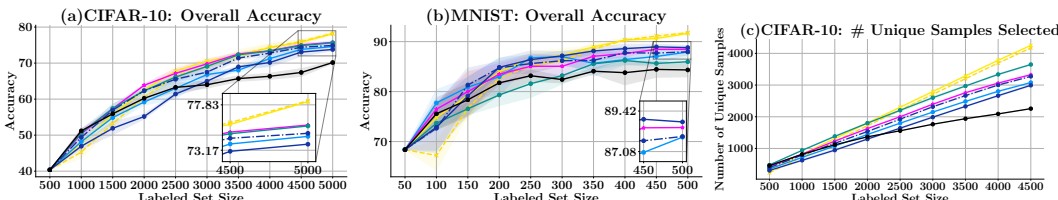

Figure 4: Active Learning under $10\times$ redundancy for CIFAR-10 and MNIST. The CG functions (LOGDETCG, FLCG) pick more unique points and outperform existing algorithms including BADGE.

of 1000 classes for imbalance and $\rho = 5$ such that $|\mathcal{C}_\mathcal{L}| + |\mathcal{D}_\mathcal{L}| = 102K$, $|\mathcal{C}_\mathcal{U}| + |\mathcal{D}_\mathcal{U}| = 664K$, $B = 25K$ and, $|\mathcal{R}| = 2.5K$. These data splits are chosen to simulate a low initial accuracy on the rare classes and at the same time maintain the imbalance factor in the labeled and unlabeled datasets.

**Results:** The results are shown in Fig. 3. We observe that SMI based functions not only consistently outperform uncertainty based methods (ENTROPY, LEAST-CONF and MARGIN) but also all the state-of-the-art diversity based methods (BADGE, GLISTER, CORESET) by $\approx 5 - 10\%$ in terms of overall accuracy and $\approx 10 - 18\%$ in terms of average accuracy on rare classes (see Fig. 3a, 3d, 3g). The reason for the same can be seen in Fig. 3c, 3f, 3i which illustrates that they fail to pick an adequate number of examples from the rare classes. Evidently, FLQMI and LOGDETMI which balance between diversity and relevance perform better than GCMI which only models relevance. Furthermore, DIV-GCMI which is a linear combination of GCMI and a diversity term performs consistently worse, which suggest that a naive combination of the two may not be as effective. This suggests the need of SMI based acquisitions functions (Equ. (2)) with richer modeling capabilities like FLQMI and LOGDETMI within SIMILAR. Furthermore, all SMI based functions also outperform the FISHER kernel based method when the validation set is small and realistic, *i.e.*, $|\mathcal{R}| = 25$. Since, [15] use a very large validation set in their experiments, we try their method FISHER-LV with a $40\times$ larger validation set of size 1000 (which is *not practical*) and observe a comparable performance with the SMI functions which use a small validation set. Furthermore, we see that FISHER-LV actually picks significantly larger number of rare class instances in MNIST, but yet is comparable in performance of FLQMI and LOGDETMI. This suggests that both these methods select higher quality and diverse rare class instances. We observe that the GC SMI variants( GCMI and DIV-GCMI) do not perform well on MNIST classification. Finally, we point out in the case of ImageNet, FLQMI performs the best and outperforms FLVMI and LOGDETMI – this is because we do not need to do the partition trick for FLQMI since it is already linear in time complexity. For FLVMI and LOGDETMI, we set the number of partitions $p = 50$ for ImageNet. Finally, we do a pairwise $t$-test to compare the performance of the algorithms (Appendix. D) and observe that the *SMI functions (and particularly* FLVMI *and* LOGDETMI*) statistically significantly outperform all AL baselines.*

### 4.2 Redundancy

**Custom dataset:** To simulate a realistic redundancy scenario we create a custom dataset by duplicating $20\%$ of the unlabeled dataset $10\times$. For CIFAR-10, the number of unique points in the unlabeled set $|\mathcal{U}| = 5K$, the initial labeled set $|\mathcal{L}| = 500$, $B = 500$, whereas for MNIST $|\mathcal{U}| = 500$, $|\mathcal{L}| = 50$ and $B = 50$. For MNIST, we also present the results for $5\times$ and $20\times$ in the Appendix. E.

**SCG vs Baselines:** As expected, the diversity and uncertainty based methods outperform random. Importantly, we observe that the SCG functions (FLCG and LOGDETCG) significantly outperform all baselines by $\approx 3 - 5\%$ towards the end as the conditioning gets stronger with increase in $\mathcal{L}$ (see Fig. 4a, 4b). This implies that simply relying on model parameters for diversity and/or uncertainty is not sufficient and that conditioning on the updated labeled set $\mathcal{L}$ (Equ. (3)) is required in batch active learning. In Fig. 4c we show that SCG based acquisition functions select significantly more unique data points than other baselines. We also perform a pairwise t-test (Appendix. E), to prove that the SCG functions consistently and statistically significantly outperform BADGE and other baselines.

### 4.3 Out-Of-Distribution

**Custom dataset:** We simulated a scenario where we convert the classification problem in CIFAR-10 and MNIST to a 8-class classification, where the first 8 classes represent the set $\mathcal{I}_F$ of in-distribution

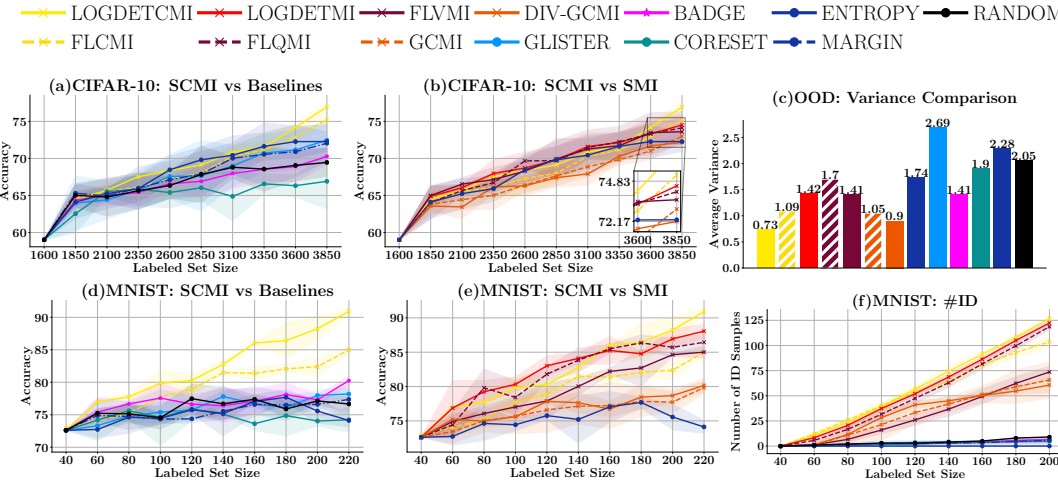

Figure 5: Active Learning with OOD data in unlabeled set. Top row: CIFAR-10 results for (a) SCMI vs Baselines, (b) SCMI vs SMI, and (c) variance comparison of different baselines, bottom row: MNIST results for (d) SCMI vs Baselines, (e) SCMI vs SMI, and (f) Number of ID points selected. We see that, i) the SCMI functions consistently outperform the baselines by $5\% - 10\%$, ii) SCMI functions outperform the corresponding SMI functions for later rounds, and (iii) SCMI functions have the least variance compared to the rest, showing that they are more robust in performance.

(ID) data points and the last 2 represent the set $\mathcal{O}_F$ of out-of-distribution(OOD) data points. The initial labeled set $\mathcal{L}$ *consists only of ID points*, i.e. $\mathcal{O}_F \cap \mathcal{L} = \emptyset$. The unlabeled set is simulated to reflect a realistic and somewhat extreme setting where the unlabeled ID data points $|\mathcal{I}_F|$ is much smaller than the unlabeled OOD data points $|\mathcal{O}_F|$. Additionally, we also assume we have a very small validation set of ID points $\mathcal{I}_V$. For CIFAR-10: $|\mathcal{L}| = 1.6K$, $|\mathcal{I}_F| = 4K$, $|\mathcal{O}_F| = 10K$, $|\mathcal{I}_V| = 40$, $B = 250$ whereas for MNIST which is a relatively simpler task, we use a smaller initial labeled sets and keep the unlabeled sets of the same size: $|\mathcal{L}| = 40$, $|\mathcal{I}_F| = 400$, $|\mathcal{O}_F| = 10K$, $|\mathcal{I}_V| = 16$, $B = 20$. Recall that our algorithm uses ID set $\mathcal{I}$ (initialized to $\mathcal{I}_V$) and OOD set $\mathcal{O}$ which we build as follows. Every time our selection approach selects a set $\mathcal{A}$, we update $\mathcal{I} = \mathcal{I} \cup (\mathcal{A} \cap \mathcal{I}_F)$ and $\mathcal{O} = \mathcal{O} \cup (\mathcal{A} \cap \mathcal{O}_F)$, i.e. we augment the ID and OOD points in $\mathcal{A}$ to the sets $\mathcal{I}$ and $\mathcal{O}$ respectively.

**SCMI vs Baselines:** Since we care about the predictive performance of the ID classes, we report the ID classes accuracy. We see that SCMI based acquisition functions significantly outperform existing AL approaches by $\approx 5 - 10\%$ (see Fig. 5a, 5d). We also observe that existing acquisition functions have a high variance, which is undesirable in real-world deployment scenarios where deep models are being continuously developed. Our SCMI based acquisition functions (LOGDETCMI and FLCMI) show the lowest variance in training (see Fig. 5c). This reinforces the need of having a framework like SIMILAR that facilitates query and conditioning sets.

**SCMI vs SMI:** We compare SCMI functions against SMI functions to study the effect of conditioning and observe that the SCMI functions are comparable to the SMI functions initially but in the later selection rounds of active learning, the SCMI functions consistently outperform SMI functions. In particular, we see an improvement of $2 - 3\%$ as the conditioning becomes stronger (see Fig. 5b, 5e). We also observe the SCMI tends to select more ID points than SMI and other baselines (see Fig. 5f), and SCMI functions have a lower variance overall compared to even the SMI functions (Fig. 5c).

## 5 Conclusion

In this paper, we proposed a unified active learning framework SIMILAR using the submodular information functions. We showed the applicability of the framework in three realistic scenarios for active learning, namely rare classes, redundancy, and out of distribution data. In each case, we observed that the functions in SIMILAR significantly outperform existing baselines in each of these tasks. Our real-world experiments on MNIST, CIFAR-10, and ImageNet show that many of the SIM functions (specifically the LOGDET and FL variants) yield $\approx 5\% - 18\%$ gain compared to existing baselines, particularly in the rare class scenario and $\approx 5\% - 10\%$ OOD scenarios. The main limitations of our work is the dependence on good representations to compute similarity. A potential negative societal impact of this work is the use of SIMILAR to perpetuate certain biases through a malicious use of the query and conditioning set. We discuss this in more detail in Appendix. G.

## Acknowledgments and Disclosure of Funding

This work is supported by the National Science Foundation under Grant No. IIS-2106937, a startup grant from UT Dallas, and by a Google and Adobe research award. Any opinions, findings, and conclusions or recommendations expressed in this material are those of the authors and do not necessarily reflect the views of the National Science Foundation, Google or Adobe.

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
