# Supplementary Material for SIMILAR: Submodular Information Measures Based Active Learning In Realistic Scenarios

## Table of Contents

# A Computational Aspects of SIM Functions in SIMILAR

## A.1 Computational complexity for selection using each function in SMI and baselines

Below, we provide a detailed analysis of the complexity of creating and optimizing the different SIM functions. Denote $|\mathcal{X}|$ as the size of set $\mathcal{X}$. Also, let $|\mathcal{U}| = n$ (the ground set size, which is the size of the unlabeled set in this case). In the main paper, we provided the high-level intuition of the complexity, ignoring the terms of $|\mathcal{P}|$ and $|\mathcal{Q}|$ since they would be typically much smaller than the number of unlabeled points $n$. For completeness, we provide the detailed complexity below:

- **Facility Location:** We start with FLVMI. The complexity of creating the kernel matrix is $O(n^2)$. The complexity of optimizing it is $\tilde{O}(n^2)$ (using memoization [18])[2] if we use the stochastic greedy algorithm [33] and $O(n^2 k)$ with the naive greedy algorithm. The overall complexity is $\tilde{O}(n^2)$. For FLQMI, the cost of creating the kernel matrix is $O(n|\mathcal{Q}|)$, and the cost of optimization is also $\tilde{O}(n|\mathcal{Q}|)$ (with naive greedy, it is $O(nB|\mathcal{Q}|)$). The complexity of FLCG is $O([n + |\mathcal{P}|]^2)$ to compute the kernel matrix and $\tilde{O}(n^2)$ for optimizing (using the stochastic greedy algorithm). Finally, for FLCMI, the complexity of computing the kernel matrix is $O([n + |\mathcal{Q}| + |\mathcal{P}|]^2)$, and the complexity of optimization is $\tilde{O}(n^2)$.

- **Log-Determinant:** We start with LogDetMI. The complexity of the kernel matrix computation (and storage) is $O(n^2)$. The complexity of optimizing the LogDet function using the stochastic greedy algorithm is $\tilde{O}(B^2 n)$, so the overall complexity is $\tilde{O}(n^2 + B^2 n)$. For LogDetCG, the complexity of computing the matrix is $O([n + |\mathcal{P}|]^2$, and the complexity of optimization is $\tilde{O}([B + |\mathcal{P}|]^2 n)$. For the LogDetCMI function, the complexity of computing the matrix is $O([n+|\mathcal{P}|+|\mathcal{Q}|]^2$, and the complexity of optimization is $\tilde{O}([B + |\mathcal{P}| + |\mathcal{Q}|]^2 n)$.

- **Graph-Cut:** Finally, we study GC functions. For GCMI, we require a $O(n|\mathcal{Q}|)$ kernel matrix, and the complexity of the stochastic greedy algorithm is also $\tilde{O}(n|\mathcal{Q}|)$. Finally, for GCCG, the complexity of creating the kernel matrix is $O(n|^2 + n|\mathcal{P}|)$, and the complexity of the stochastic greedy algorithm is $\tilde{O}(n^2 + n|\mathcal{P}|)$.

We end with a few comments. First, most of the complexity analysis above is with the stochastic greedy algorithm [33]. If we use the naive or lazy greedy algorithm, the worst-case complexity is a factor $B$ larger. Secondly, we ignore log-factors in the complexity of stochastic greedy since the complexity is actually $O(n \log 1/\epsilon)$, which achieves an $1 - 1/e - \epsilon$ approximation. Finally, the complexity of optimizing and constructing the FL, LogDet, and GC functions can be obtained from the CG versions by setting $\mathcal{P} = \emptyset$.

## A.2 Details on Partitioning Approach

In some of our experiments, we choose to partition the unlabeled set into chunks in order to meet the scale of the dataset used in that experiment. This is because many of the techniques (specifically LogDet functions, FLVMI, FLCG, FLCMI, GCCG) all have $O(n^2)$ space complexity. For $n$ in the range of a few million to a few billion data points (which is not uncommon in big-data applications today), we need to scale our algorithms to be linear in $n$ and not quadratic. For this, we propose a simple partitioning approach where the unlabeled data is chunked into $p$ partitions. In this strategy, we perform unlabeled instance acquisition on each chunk using a proportional fraction of the full AL batch size. The most notable example of the use of our partitioning strategy is in our down-sampled ImageNet experiment. By performing AL acquisition on the full unlabeled set, almost all AL strategies exhaust the available compute resources. Hence, to execute most of our AL strategies, we partitioned the unlabeled set into 50 equally sized chunks, so each partition has around 10k to 20k instances. As $n$ grows, the number of partitions would also grow so that $n/p$ is roughly constant. The complexity of most approaches discussed above would then be $O(n^2/p)$ ($O(n^2/p^2)$ for each chunk, repeated $p$ times), and if $n/p = r$ is a constant, then the complexity $O(nr)$ would be linear in $n$. We then acquire a number of unlabeled instances from each chunk whose ratio with the full AL batch size is equal to the ratio between the chunk size and the full unlabeled set. The acquired instances from each chunk are then combined to form the full acquired set of unlabeled instances.

---

[2]$\tilde{O}$: Ignoring log-factors

# B  More Details on Experimental Setup, Datasets, and Baselines

## B.1  Datasets description in each scenario

We used various standard datasets – namely, MNIST, CIFAR10, and ImageNet – to demonstrate the effectiveness and robustness of SIMILAR. We also provide additional experiments on SVHN in sections below. We use standard sources for all datasets. As previously mentioned, we perform our experiments on a down-sampled version of ImageNet. Beyond the fact that each image is now $32 \times 32$, the data set is otherwise identical. Moreover, we find that the provided validation set is often used as the test set in most evaluations on down-sampled ImageNet. The down-sampled ImageNet training set can be procured here, and the validation set can be found here. Note that associated licenses for all datasets apply.

**Rare classes setting:**    In Tab. 4, we show the exact initial splits used in our experiments for the rare classes scenario. In CIFAR-10, ImageNet, and SVHN, we use randomly chose half the number of classes as imbalanced and the other half as balanced. Following [15], we chose classes $(5, \cdots 9)$ as imbalanced classes in MNIST. We use an AL batch size of 125 for the CIFAR-10, MNIST and SVHN datasets. We use the same data setting for the CIFAR-10 and SVHN datasets with an imbalance factor $\rho = 20$. The results for SVHN are in Appendix. D. For MNIST, we additionally show results for $\rho = 100$ in Appendix. D. Due to the scale of down-sampled ImageNet and the natural imbalance present in its full training set, we adopt a different dataset splitting strategy. Following [15], we randomly chose 500 classes (half) as rare classes. Our train set is initialized as having 34 examples per rare class and 170 examples per normal class. Our validation set contains 5 examples per class, making it balanced. The unlabeled set is created to have 1 rare example for every 5 normal examples. In all, our initialization leads our initial train set, validation set, and unlabeled set to have approximately 100k, 5k, and 660k points, respectively. We use an AL batch size of 25k points, and we use the same training conditions as before. However, we perform AL selection by dividing the unlabeled set into chunks (partitions), selecting a proportionate fraction of the AL batch size from each. In this case, we divide the unlabeled set into 50 to 100 partitions (determined by compute limitations) and perform selection on each partition.

| Dataset | Imbalance factor ($\rho$) | Labeled (per class) | Valid (per class) | Unlabeled (per class) |
|---|---|---|---|---|
| CIFAR-10 | 20 | 3 | 5 | 150 |
| SVHN | | 22 | 5 | 3000 |
| MNIST | 20 | 3 | 5 | 200 |
| | | 22 | 5 | 4000 |
| | 100 | 3 | 5 | 40 |
| | | 22 | 5 | 4000 |

Table 4: Number of data points for each dataset in the rare classes scenario. For CIFAR-10, MNIST, and SVHN, we use 5 balanced classes and 5 imbalanced classes. In the main paper, we show experiments for $\rho = 20$. In Appendix. D, we show experiments for $\rho = 100$.

**Redundancy setting:**    In Tab. 5, we show the exact initial splits used in our experiments for the redundancy scenario. For CIFAR-10 and SVHN, we use the same setting. Since MNIST classification is a relatively simpler problem, we use one tenth of the data points used in the CIFAR-10 setting. For all datasets, we create the unlabeled dataset by duplicating $20\%$ of the unlabeled dataset RF $\times$. We denote RF as the redundancy factor. For instance, we consider 5000 unique points and duplicate $20\%$ of them $10\times$ in CIFAR-10. This gives us $(5000 \times 0.2 \times 10 = 10000)$ duplicated points and $(5000 - (5000 \times 0.2 \times 10) = 4000)$ original points for a total of $(10000 + 4000 = 14000)$ points.

| Dataset | Total Unique Points | Fraction of points duplicated | Number of duplicated points |
|---|---|---|---|
| CIFAR-10, SVHN | 5000 | 20% | 5000*0.2*RF |
| MNIST | 500 | 20% | 500*0.2*RF |

Table 5: Number of data points for each dataset in the redundancy scenario. RF here is the redundancy factor. In the main paper, we show experiments for RF=$10\times$. In Appendix. E, we show experiments for RF=$5\times$ and RF=$20\times$.

**Out-of-distribution setting:** In Tab. 6, we show the exact initial splits used in our experiments for the out-of-distribution scenario. In all datasets, we chose the first 8 classes to be in-distribution (ID) and the last 2 classes to be out-of-distribution (OOD). Initially, the labeled set consists of only ID points. The unlabeled set is designed to reflect a realistic setting with high number of OOD points. For CIFAR-10, we use 200 points per ID class in the labeled set and 500 points per ID class, 5000 points per OOD class in the unlabeled set. This gives us an initial labeled set of size $200 \times 8 = 1600$ and an initial unlabeled set of size $500 \times 8 + 5000 \times 2 = 14000$. We make the task slightly challenging for MNIST by further decreasing the number of ID points in the unlabeled dataset as shown in Tab. 6.

| Dataset | | Labeled (per class) | Valid (per class) | Unlabeled (per class) |
|---|---|---|---|---|
| CIFAR-10 | ID points | 200 | 5 | 500 |
| | OOD points | 0 | 0 | 5000 |
| MNIST | ID points | 5 | 2 | 50 |
| | OOD points | 0 | 0 | 5000 |

Table 6: Number of data points for each dataset in the out-of-distribution scenario.

## B.2 Experimental setup

We ran experiments using an SGD optimizer with an initial learning rate of 0.01, a momentum of 0.9, and a weight decay of 5e-4. We decay the learning rate via cosine annealing [31] for each epoch. For MNIST, we use the LeNet model [29]. For all other datasets, we use ResNet18 model [17]. For each round of active learning, we train until the accuracy reaches 99% or the epoch count reaches 150. We run all our experiments on a single V100 GPU.

## B.3 Details on computation of penalty matrix

The penalty matrices computed in this paper follow the strategy used in [3]. In their strategy, a penalty matrix is constructed for each dataset-model pair. Each cell $(i, j)$ of the matrix reflects the fraction of training rounds that AL with selection algorithm $i$ has higher test accuracy than AL with selection algorithm $j$ with statistical significance. As such, the average difference between the test accuracies of $i$ and $j$ and the standard error of that difference are computed for each training round. A two-tailed $t$-test is then performed for each training round: If $t > t_\alpha$, then $\frac{1}{N_{train}}$ is added to cell $(i, j)$. If $t < -t_\alpha$, then $\frac{1}{N_{train}}$ is added to cell $(j, i)$. Hence, the full penalty matrix gives a holistic understanding of how each selection algorithm compares against the others: A row with mostly high values signals that the associated selection algorithm performs better than the others; however, a column with mostly high values signals that the associated selection algorithm performs worse than the others. As a final note, [3] takes an additional step where they consolidate the matrices for each dataset-model pair into one matrix by taking the sum across these matrices, giving a summary of the AL performance for their entire paper that is fairly weighted to each experiment. We present the penalty matrices for each of the settings in the sections below.

## B.4 Licensing details

**Datasets.** Our experiments with SIMILAR utilize the following datasets.

- CIFAR-10 [27]: MIT License
- MNIST [30]: Creative Commons Attribution-Share Alike 3.0
- SVHN [35]: CC0 1.0 Public Domain
- ImageNet [38]: Custom (Research, Non-Commercial)

**Repositories.** Our experiments utilize contributions from existing code repositories. Specifically, we utilize the DISTIL repository for AL baselines. We utilize the Fisher Kernel Self-Supervision repository in our usages of FISHER and its variants. We extensively use PyTorch, and we utilize the CORDS repository in our gradient computations. To summarize, the following repositories are used, and their licenses from their original sources are also provided:

- PyTorch [36]: Modified BSD

- DISTIL: MIT License
- CORDS: MIT License
- Fisher Kernel Self-Supervision [15]: (None Listed)
- BADGE [3]: None Listed

### B.5 Baselines and Code

For all baselines, we use code either from existing libraries and codebases or from the authors. For BADGE [3], we use the code from the authors[3]. Similarly, for the FISHER baseline, we use the code from the authors[4]. For the other methods like entropy sampling, CORESET, etc., we use DISTIL[5], which implements most of the state-of-the-art standard AL approaches building upon the respective authors code.

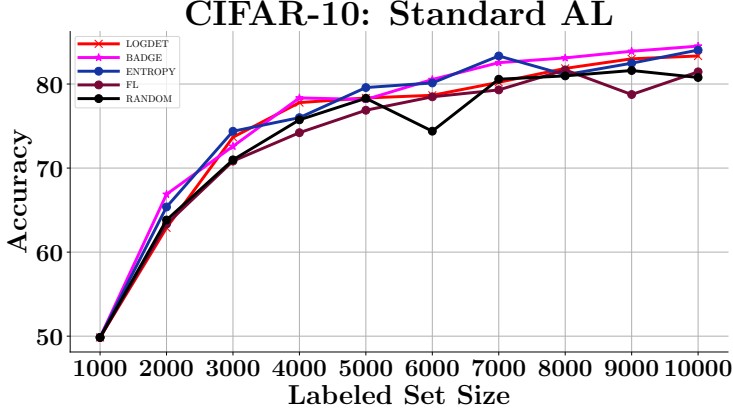

Figure 6: Comparison of submodular functions with baselines in a standard active learning setting.

## C   Results with Standard Active Learning

In Figure 6, we compare the performance of the SFs on standard AL – i.e., without redundancy, out-of-distribution data, and imbalance. The basic idea here is that we compute the similarity kernels using the gradients of the model (Algorithm 1) and use just the submodular function – i.e., setting $\mathcal{Q} = \mathcal{U}, \mathcal{P} = \emptyset$. In this work, we use the log-determinant and the facility location functions. We make the following observations: **1)** Log-determinant functions perform comparable to BADGE and entropy sampling, particularly in the beginning. **2)** The facility location function does not perform as well in the standard AL setting, implying that diversity tends to play a more important role in standard active learning compared to representation.

## D   Additional Experiments and Takeaways for Active Learning with Rare Classes

In Figure 7, we show additional results for MNIST and SVHN for active learning with rare classes. The top row shows the results for the extreme imbalance scenario, i.e., $\rho = 100, B = 25$ (small batch size and extreme imbalance). We observe that LOGDETMI significantly outperforms all other techniques, and FLQMI and FISHER come next. Note that the FISHER baseline [15] was originally presented in this extreme imbalance scenario. The middle row in Figure 7 contains results for $\rho = 20, B = 25$. This is similar to the results presented in the main paper but using a much smaller batch size. Here, LOGDETMI and FLQMI again outperform the other baselines. While the average performance of the FISHER baseline [15] is comparable to LOGDETMI and FLQMI, it has a much higher variance compared to others (Figure 8). Finally, the bottom row shows the performance of

---

[3]https://github.com/JordanAsh/badge
[4]https://github.com/gudovskiy/al-fk-self-supervision
[5]https://github.com/decile-team/distil

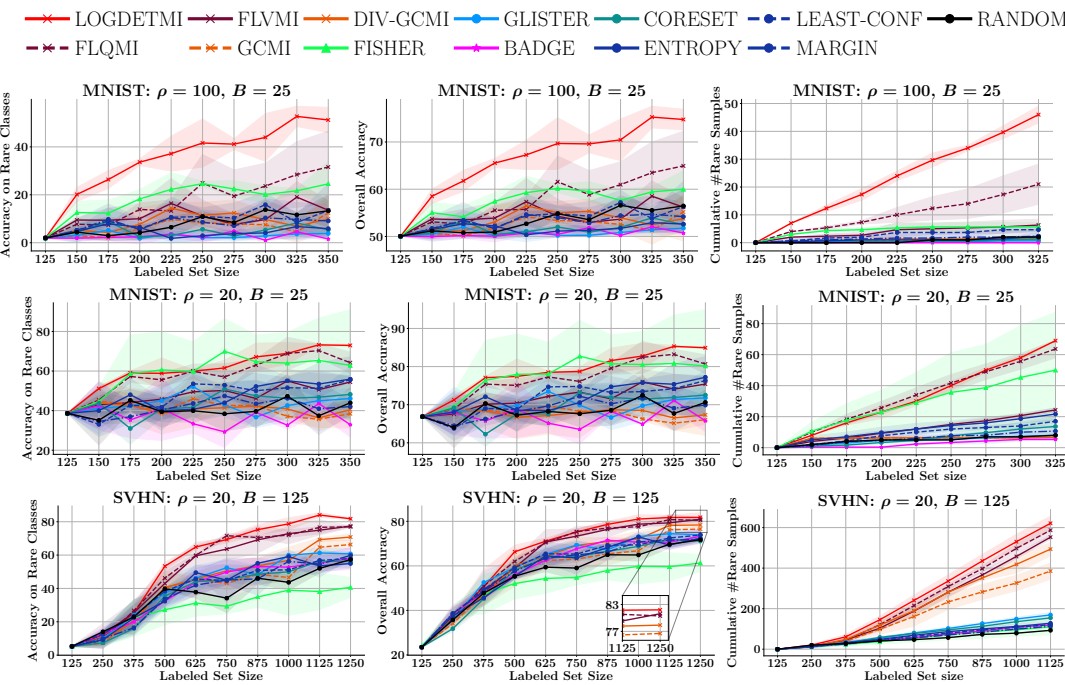

Figure 7: Additional experiments on MNIST and SVHN for active learning with rare classes. **Top row:** MNIST $\rho = 100, B = 25$, LOGDETMI outperforms other methods even in extreme imbalance, with a large gap in accuracy, followed by FLQMI. **Middle row:** MNIST $\rho = 20, B = 25$, LOGDETMI and FLQMI outperform all baselines in the later rounds of AL. **Bottom row:** SVHN $\rho = 20, B = 125$, All SMI methods significantly outperform other baselines.

Figure 8: Variance comparison on the rare classes scenario for MNIST $\rho = 20, B = 25$ (Middle row in Fig. 7). FISHER has $\approx 5\times$ variance in comparison with the SMI methods. The figure shares the same legend as Fig. 7.

the different techniques on SVHN. Again, we see that LOGDETMI and FLQMI outperform all other techniques.

**Takeaways from the Results:** The following are the main takeaways of the experiments in this section and the main paper:

- Among the different MI functions, LOGDETMI and FLQMI outperform all other MI functions. They also mostly outperform the Fisher Kernel baseline which was also designed for dealing with rare classes [15].

- LOGDETMI particularly outperforms every other method in the high imbalance regime (100x imbalance). This is mainly because it is able to select the highest number of points from the rare classes (top row, right most plot in Figure 7.

- The FISHER baseline also can have a high variance, particularly when the batch size is high.

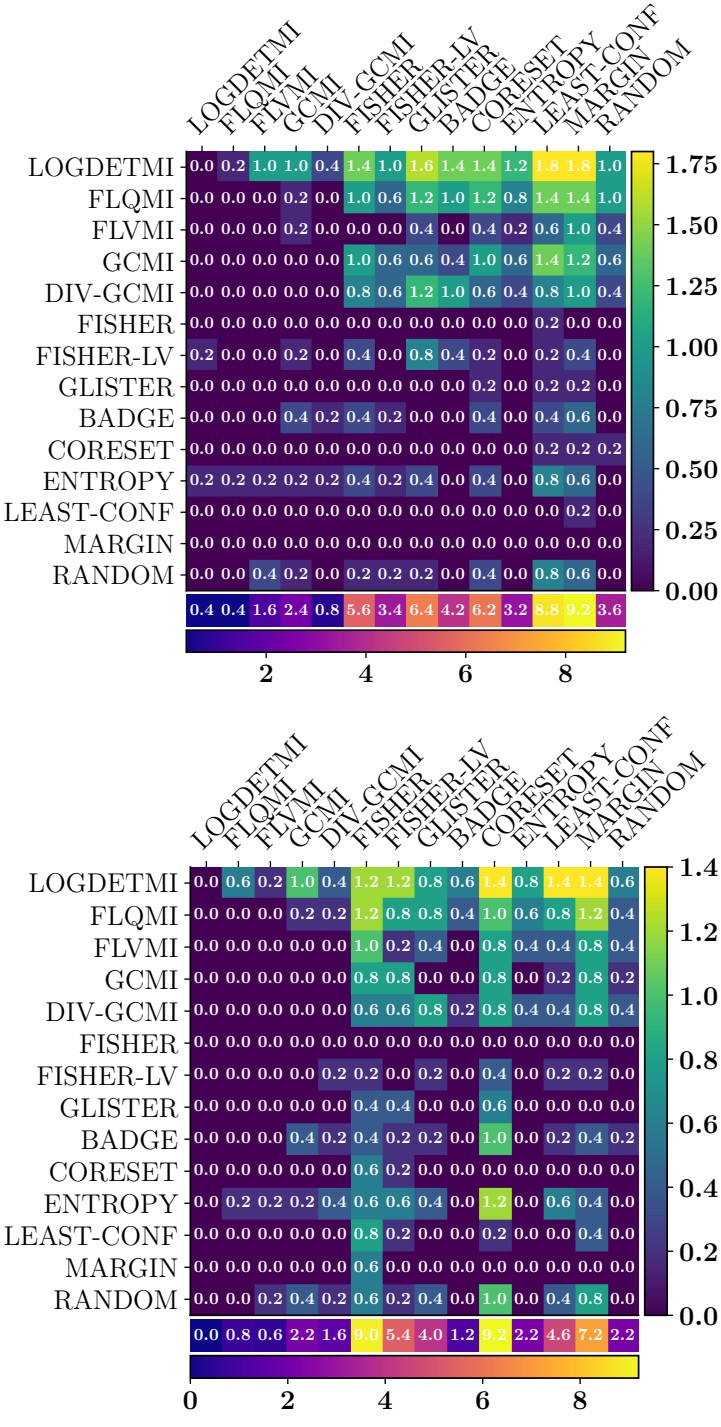

Figure 9: Penalty Matrix comparing the average accuracy of rare classes (**top**) and overall accuracy (**bottom**) of different AL approaches in the class imbalance scenario. We observe that the SMI functions have a much lower column sum compared to other approaches.

- For a fair comparison, we used a very small validation set in all our experiments. As compared in the main paper, FISHER performance does improve when we use a larger validation set, but doing so is not realistic.

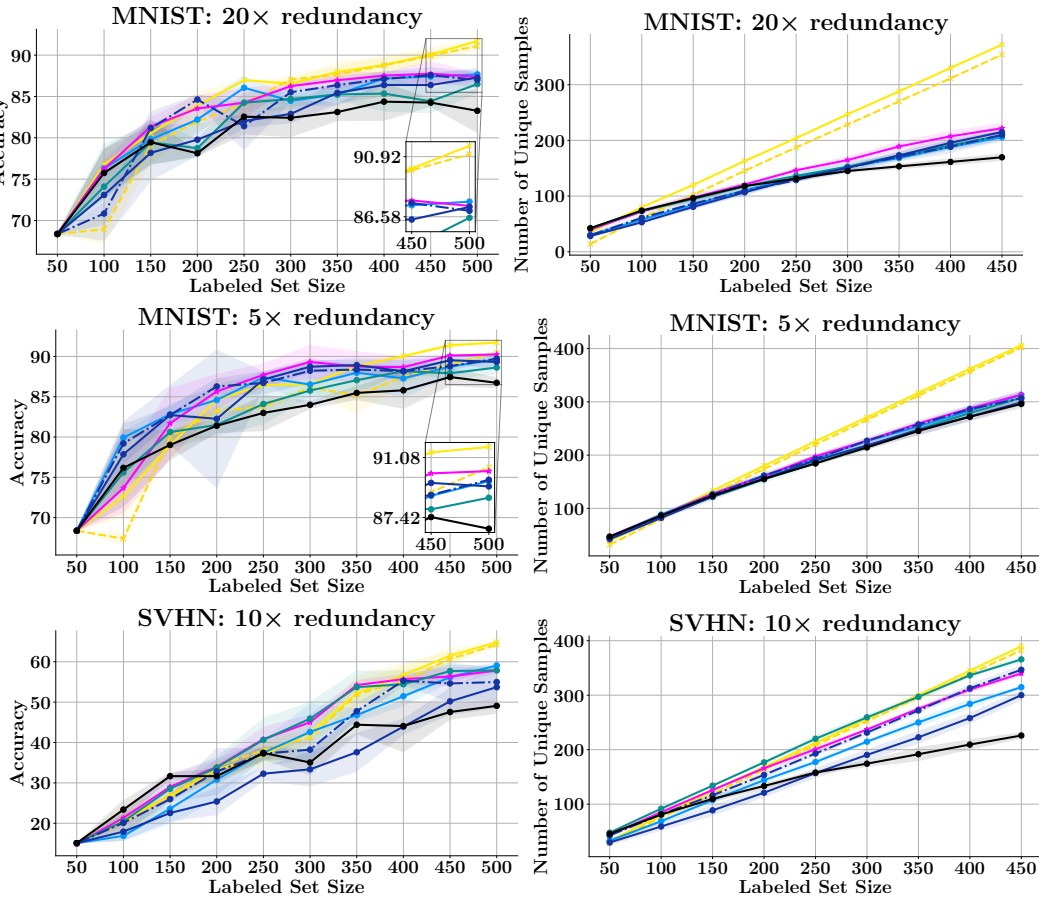

Figure 10: Active Learning under $20\times$ redundancy (**top row**) and $5\times$ redundancy (**middle row**) on MNIST. **Bottom row:** $10\times$ redundancy on SVHN. The CG functions (LOGDETCG, FLCG) pick more unique points and outperform existing algorithms including BADGE.

- FLQMI is more scalable compared to LOGDETMI and other kernel-based approaches; hence, it is the desired choice of approach for very large datasets.

**Penalty Matrix:** Figures 9 shows the penalty matrix results on the rare class accuracy (top) and overall accuracy (bottom). We see that LOGDETMI and FLQMI have the smallest column sum, which indicates that most other baselines are not statistically significantly better than them. Furthermore, they also have the highest row sum (followed by some of the other MI functions), which indicates that they are statistically significantly better than other approaches. These matrices are obtained by combining the results on MNIST and CIFAR-10 for $\rho = 20, B = 125$ (i.e., the results in the main paper).

# E  Additional Experiments and Takeaways from Active Learning with Redundancy

In the main paper, we show the results on CIFAR-10 and MNIST with $10\times$ redundancy. In this section, we also add results for $5\times$ and $15\times$ redundancy for MNIST. The results are in Figure 10. Furthermore, we also run experiments on SVHN (bottom row) with $10\times$ redundancy. The following are the takeaways of the results:

- The CG functions (LOGDETCG and FLCG) significantly outperform other baselines including BADGE, particularly after a few rounds of AL and towards the end. In particular, there is a

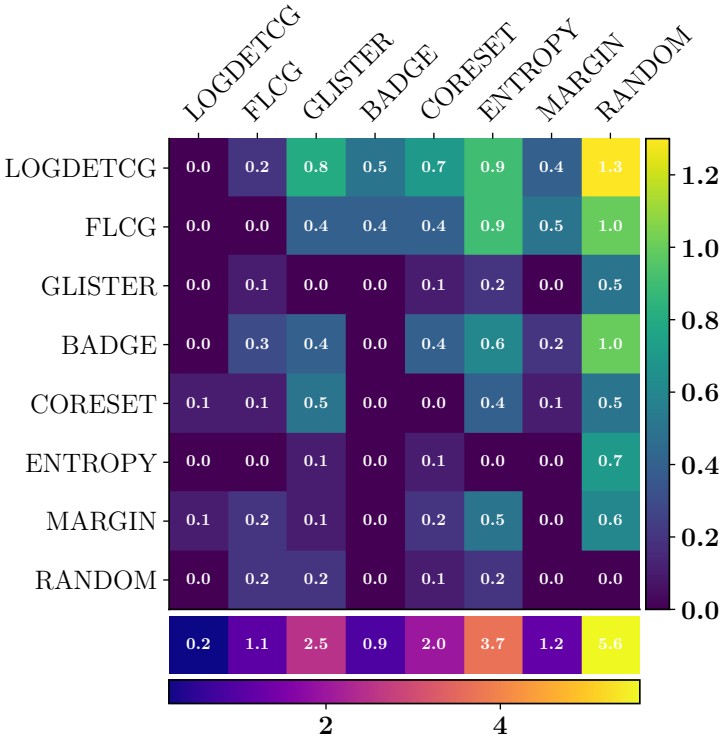

Figure 11: Penalty Matrix comparing the different AL approaches in the redundancy scenario. We observe that the SCG functions have a much lower column sum compared to other approaches.

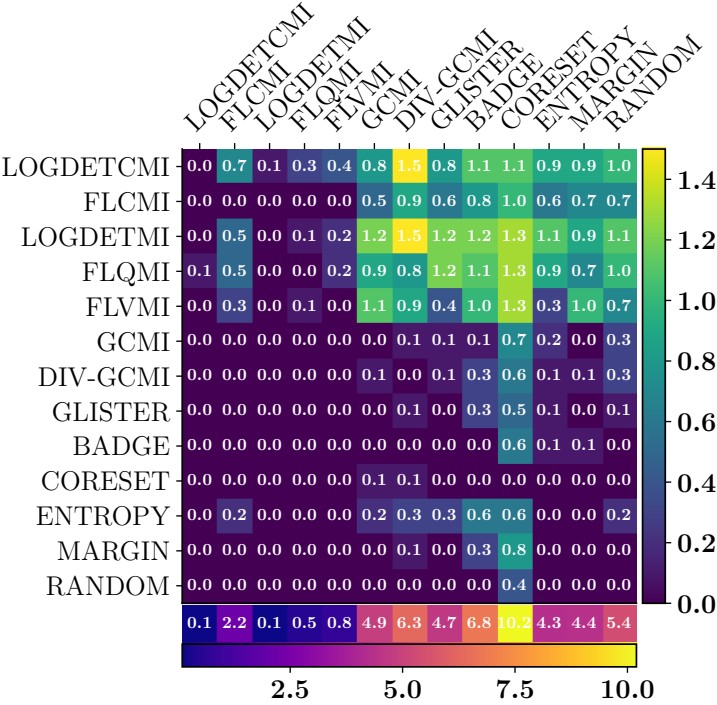

Figure 12: Penalty Matrix comparing the different AL approaches in the OOD Scenario. We observe that the SCMI and SMI functions have a much lower column sum compared to other approaches.

improvement of 3% to 5% using the CG functions compared to BADGE and other baselines with a labeled set size of 500.

- The main reason for this is that the CG functions pick more unique points compared to the other techniques.
- Amongst the two CG functions, we see that LOGDETCG performs better than FLCG.
- From the pairwise penalty matrix in Figure 11, we see that LOGDETCG has the lowest column sum and has the highest row sum, which indicates that it statistically significantly outperforms other techniques. In terms of the row sum, LOGDETCG is followed by FLCG and BADGE.

## F   Additional Experiments and Takeaways for Active Learning with OOD Data

In the case of active learning with OOD data, we additionally add the penalty matrix (figure 12). The following are the main observations and takeaways:

- Figure 12 shows the results of the penalty matrix with the different CMI functions. We observe that LOGDETCMI has the smallest column sum along with LOGDETMI.
- However, as shown in the main paper, the CMI functions have the smallest variance and are hence more stable compared to the SMI variants. Furthermore, the CMI functions generally outperform the SMI counterparts at later rounds.
- However, the SMI functions are often comparable (particularly LOGDETMI and FLQMI) and hence are a good choice for OOD data as well.

## G   Societal Impacts and Limitations

**Limitations of this work:**   The first limitation of this work is that the MI functions are all graph-based functions. With the exception of FLQMI, all functions have quadratic complexity. The partitioning trick will help, but that comes at the cost of performance. We would like to explore more classes of MI functions (feature-based functions [45] in particular) in future work. Secondly, the MI functions depend on good choices of features. In this work, we use gradients which tend to work very well since they inherently also capture uncertainty [3]. However, the approaches do not perform as well in the early stages, which could be mitigated by the use better features, e.g., self-supervised and unsupervised representations [15].

**Societal Impacts:**   Negative societal impacts of this work include using SIMILAR to mine through large datasets to perpetuate and amplify certain biases in the data. On the flip side, this work can also have a positive impact through its use for fair active learning, where certain under-represented and minority slices or classes can be improved upon by applying it in the rare class and rare slice experiment setting (Sec. 3.2). We would like to explore the use of SIMILAR in applications like improving the performance of biased slices based on race; for example, we would like to improve inference performance on underrepresented Asian woman using SIMILAR for tasks like face recognition, gender recognition, and age recognition. Importantly, recent work has shown that commercial facial recognition and age/gender classification engines perform poorly on these rare slices [7]. A number of recent papers have been proposed to generate such fair face datasets [21], but creating such datasets can take a lot of manual effort to mine the rare slices. We propose to use and study SIMILAR for such scenarios in future work.

## H   Experiments on Real-world Medical Dataset

In this section, we apply our framework to Pneumonia-MNIST (pediatric chest X-ray) medical image classification dataset. The goal is to classify X-ray images into 'pneumonia' and 'benign'. As done in Sec. 4.1 and to simulate a real-world scenario, we use an imbalance factor $\rho = 20$, such that the 'pneumonia' class is a rare class. We use $|\mathcal{C}_\mathcal{L}| + |\mathcal{D}_\mathcal{L}| = 105$, $|\mathcal{C}_\mathcal{U}| + |\mathcal{D}_\mathcal{U}| = 1100$, $B = 10$ (AL batch size) and, $|\mathcal{R}| = 5$. On this dataset, we observed that using misclassified data points in $\mathcal{R}$ is beneficial for acquiring subsets that lead to higher accuracy gains. We observe that the SMI functions outperform the baselines by $\approx 10\% - 12\%$ on the rare classes accuracy and $\approx 8\% - 10\%$ on the overall accuracy (see Fig. 13).

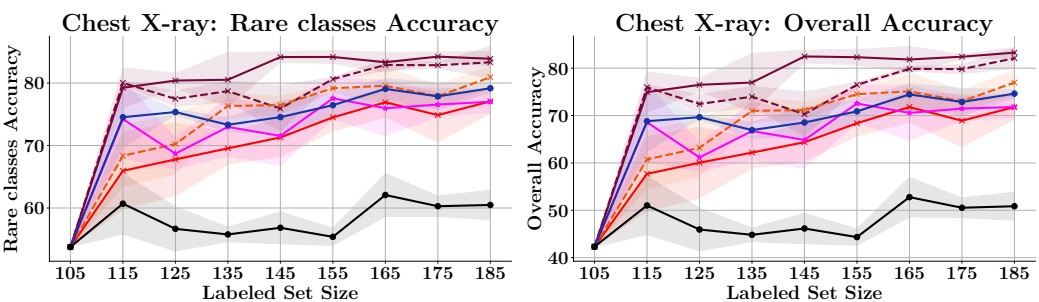

## Chest X-ray: Rare classes Accuracy

## Chest X-ray: Overall Accuracy

Figure 13: Active Learning on real-world medical image classification. The SMI functions outperform the baselines by $\approx 10\% - 12\%$ on the rare classes accuracy and $\approx 8\% - 10\%$ on the overall accuracy.

# I Experiments on Multiple Realistic Scenarios

In this section, we apply our framework to a scenario where redundancy *and* rare classes are co-occurring in the dataset. To do so, we first create an imbalance on CIFAR-10 in a similar fashion as done in Sec. 4.1. We use $\rho = 10, |\mathcal{C}_\mathcal{L}| + |\mathcal{D}_\mathcal{L}| = 125, |\mathcal{C}_\mathcal{U}| + |\mathcal{D}_\mathcal{U}| = 5.5K, B = 100$ and repeat the unlabeled dataset $5\times$ to get $|\mathcal{C}_\mathcal{U}| + |\mathcal{D}_\mathcal{U}| = 27.5K$. We observe that the SCMI and SMI functions perform better than the baselines (see Fig. 14).

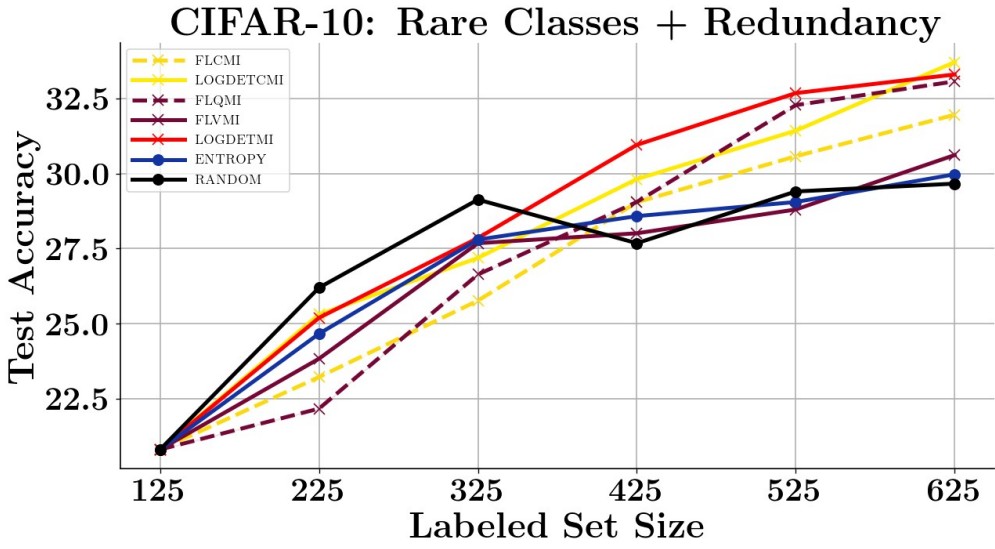

Figure 14: Active learning in multiple realistic scenarios (Rare classes + Redundancy).