# OpenReview forum: "SIMILAR: Submodular Information Measures Based Active Learning In Realistic Scenarios"
_NeurIPS.cc/2021/Conference — NeurIPS 2021 Poster_

### Official Review · Reviewer_ryua · 2021-07-05

**Rating:** 5
**Confidence:** 3

**Summary:**

In this paper, the authors propose methods for selecting a query set for batch active learning. These methods select a query set by maximizing a criterion based on submodular functions such as facility location function, graph cut function, and log-determinant function. The similarity matrix between data points, used in the definition of submodular functions, is defined by the inner product of the gradients of loss functions.

The authors empirically validate the efficiency of the proposed methods for batch active learning with rare classes, redundancy, and out-of-distribution data points. They adopt standard image datasets and standard deep neural network models. In each experimental setting, some of the proposed methods outperform existing methods for batch active learning.

**Ethical Concerns:**

There is no ethical concern.

**Limitations And Societal Impact:**

The discussion on the limitations and societal impact is adequate.

**Main Review:**

The overall writing quality is very well. The main body is well organized, and all the descriptions are concise. The experiments are conducted thoroughly on standard datasets and models, which validate the superiority of the proposed methods.

The main contribution of this paper is criteria based on submodular functions for selecting a query set, but I do not fully support their originality and significance. The authors proposed three criteria: submodular mutual information (SMI), submodular conditional gain (SCG), and submodular conditional mutual information (SCMI). Applying these criteria to batch active learning does not seem to be completely novel. SCG is a standard criterion when a submodular utility function is used for batch active learning (e.g., Chen and Krause, 2013, Fern et al., 2017). SMI was proposed in previous work (Iyer et al., 2021). SCMI is just a combination of these two. Also, SMI and SCMI do not seem to satisfy submodularity, and then it does not seem easy to optimize it with theoretical guarantees. It loses the most prominent advantages of using submodular functions.

---

I read the authors' response. As the authors claimed, the novel part of this paper is to apply SMI and SCMI to batch active learning, but submodular optimization techniques cannot be used for these criteria except in some special cases. I am not fully convinced by the merit of the proposed framework, so I keep my score.

**Time Spent Reviewing:**

5

---

> ### Author Response · Authors · 2021-08-10
> **Response to Reviewer ryua**
>
> Thank you very much for taking the time to review our submission and providing your valuable comments! We’re glad that you enjoyed the writing of our submission. Here, we address some points made in your discussion.
>
> > Q1: “Applying these criteria to batch active learning does not seem to be completely novel. SCG is a standard criterion when a submodular utility function is used for batch active learning (e.g., Chen and Krause, 2013, Fern et al., 2017). SMI was proposed in previous work (Iyer et al., 2021). SCMI is just a combination of these two.”
>
> A1: While the notion of SMI and SCMI functions are not novel contributions of this work, the SIMILAR framework which uses these measures is novel. Namely, our main focus is to present the applicability of submodular information measures for targeted batch active learning. We show that SIMs provide a common framework for approaching various realistic AL scenarios, such as OOD, rare classes, and redundant data. While examples of SIMs can be found in the AL literature (especially SCG as you’ve pointed out), our introduction of SIMILAR provides a framework that encapsulates the use of existing SIMs and provides further insight to their applicability for AL.
>
> > Q2: “Also, SMI and SCMI do not seem to satisfy submodularity, and then it does not seem easy to optimize it with theoretical guarantees. It loses the most prominent advantages of using submodular functions.”
>
> A2: While it is true that not all differences of submodular functions are submodular, (Iyer et al., 2021) actually show that, under certain conditions, some SMI functions $I_f(A;B)$ are submodular given fixed $B$ under their Thm. 5 -- specifically, the FLMI variants and GCMI functions are in fact submodular. While not all submodular instantiations for $f$ will guarantee submodular $I_f(A;B)$, we can retain the theoretical guarantees for the cases when it is submodular. Studying the (weak) submodularity and theoretical properties of the functions which are not submodular will be left for future work.

---

### Official Review · Reviewer_FgsV · 2021-07-07

**Rating:** 5
**Confidence:** 3

**Summary:**

This paper proposes a "unified active learning framework". The key element of this framework is a submodular information measure, denoted by $I_f(A;Q|P)$ where $f$ is a submodular score function, $A$ is a set of unlabeled samples whose "value" is measured, and $P$ and $Q$ are set of samples to be specified by the algorithm as parameters. $I_f(A;Q|P)$ is supposed to measure similarity and dissimilarity among $A$, $P$, and $Q$. In each round of the proposed "unified active learning framework", it chooses $A$ that maximizes $I_f$. By choosing $P$ and $Q$ properly (for example, for the class imbalance task, setting $Q$ to be a small labeled sample set from rare classes), this paper shows empirically the proposed algorithm outperforms other active learning algorithms on various tasks, including standard AL, AL with class imbalance, AL with redundant data, and AL with out of distribution data.

**Limitations And Societal Impact:**

See main review.

**Main Review:**

Pros:
It is nice to see a unified AL framework that can address different AL tasks. This is novel as far as I can tell and can be interesting for the community. The main tool, submodular information measure (SIM), looks also a very interesting tool.

Cons:
- My major concern is clarity, in particular the explanation of SIM and how it is used.
  - It would be much clearer if the authors could walk through at least a few instantiations of SMI in table 2: what is the corresponding f? how are they derived? How can we understand them?
  - Are there any connections between the proposed framework with any existing AL heuristics?
  - Why, in line 146, the inner product between gradients a good similarity measure? Are there any other choices and how do they compare?
- Experiment results look good, but seem unfair: comparisons are between *specialized* versions (adapted to specific learning scenarios by choosing proper parameters) of SIM vs other *general-purpose* AL algorithms. It would be interesting to see how the proposed method compares against known methods for rare classes, redundant data, out of distribution data. (for example, over/under sampling, tweaking loss the loss function).
- As admitted in Section 3.5, a single iteration would take ~n^2 time where n=the number of unlabeled samples. This would be too expensive when n is not small. The paper provides some remedies, but I'm not sure how good they are (in terms of both accuracy and time). It would be more clearer if the author could provide comparison of running time in experiments to help the readers evaluate.

Other issues:
- confidence bar
  - How are confidence bars defined and computed?
  - Related, why for many plots the error bars shrink to a point (for example,figure 3(d), DIV-GCMI at labeled set size=1000), or even worse disappear (for example, figure 3(d) LOGDETMI after 1125)?
  - Why don't we have them for image net dataset?
- It would be more convincing if you could present experiment results with different choices of hyper parameters (for example, learning rates, batch sizes, etc.)

--

I've read the author feedback and I tend to keep my ratings for its current version. I think this paper has good potential, but it would be much better to me if it could provide more explanation for the proposed framework, and make it clear what is new and connection with existing methods. Regarding Q4, I'm not very sure if the methods you mentioned are standard baselines, but still comparing with simple (and effective IMHO) heuristics would make it more convincing.

**Time Spent Reviewing:**

4

---

> ### Author Response · Authors · 2021-08-10
> **Response to Reviewer FgsV**
>
> Thank you very much for taking the time to review our submission and providing your valuable comments! We are happy to respond to your comments as below:
>
> > Q1: It would be much clearer if the authors could walk through at least a few instantiations of SMI in table 2
>
> A1: Thank you for pointing this out. We will make this clear in the next version of the paper.
>
> > Q2: Are there any connections between the proposed framework with any existing AL heuristics?
>
> A2: We do not explicitly mention this in the paper, but it turns out that the SMI functions are closely related to and generalize a recent AL approach called GLISTER-ACTIVE [a]. It is also related to two data selection approaches, CRAIG [b] and GRADMATCH [c]. We will add some details on these connections in the next version of the paper.
>
> > Q3:  Why, in line 146, the inner product between gradients a good similarity measure? Are there any other choices and how do they compare?
>
> A3:  Note that if the gradient vectors are L2-normalized, then the inner product between the gradients gives exactly the cosine similarity. Beyond cosine similarity, we also tried using negative Euclidean distance as a similarity measure. Both measures have been used in many papers [a,b,c,d,e] and performed similarly in our experiments. For consistency across all experiments in the paper, we use cosine similarity. As an ablation study, we compare the performance of FLQMI instantiated with a cosine similarity kernel with FLQMI instantiated with a negative Euclidean distance kernel. The table below summarizes the results on CIFAR-10 classification in the rare classes scenario:
>
> | Function              | Initial accuracy | R1    | R2    | R3    | R4    | R5    | R6    | R7    | R8    | R9    |
> |-----------------------|------------------|-------|-------|-------|-------|-------|-------|-------|-------|-------|
> | FLQMI (Cosine Sim)    |            21.72 | 28.98 | 31.89 | 35.11 | 37.92 | 41.54 | 42.26 | 47.96 | 47.18 | 51.87 |
> | FLQMI (Negative Dist) |            21.72 | 27.76 | 32.49 | 35.09 | 35.66 | 40.56 | 40.58 | 45.53 | 50.15 | 51.58 |
>
> > Q4:  It would be interesting to see how the proposed method compares against known methods for rare classes, redundant data, out of distribution data.
>
> A4: Figure 3 in the main paper shows our comparison against such known methods. GLISTER [a] (robust learning), FISHER [d] (AL for biased datasets) and FISHER-LV [d] are baselines that are known methods for dealing with such scenarios. Infact, the main motivation of FISHER [d] is “to consider a realistic setting for AL, where the unlabeled dataset is biased”. We can see that our methods (specifically LOGDETMI, FLQMI) outperform other baselines by more than 10% on the rare classes.
>
> > Q5: As admitted in Section 3.5, a single iteration would take ~n^2 time where n=the number of unlabeled samples. This would be too expensive when n is not small. The paper provides some remedies, but I'm not sure how good they are (in terms of both accuracy and time). It would be more clearer if the author could provide comparison of running time in experiments to help the readers evaluate.
>
> A5: Thank you for your suggestion! We will add a comparison of running time in the next version of the paper.
> As an ablation study, we analyzed the trade-off between the speedup acquired through the partition trick and the drop in accuracy. To do so, we ran experiments on CIFAR-10 with 5 partitions and without the partitioning. We ran one round of active learning and compared this trade-off for some SMI functions used in SIMILAR. We observe that with FLVMI, we have around a 0.5% accuracy loss while with LOGDETMI, we have around a 1% accuracy drop. In both cases, this results in close to a 5x speedup in the subset selection time and a 25x memory savings (since we need to compute the kernel for only 1/5th of the data points). Also, note that FLQMI does not require partitioning (see Ln. 236-237 in the main paper) since it is linear in the ground set size (since the query size is often small). In the ImageNet case, we used 50 partitions for the FLVMI and LOGDETMI functions mainly because of memory limitations. We will add a few more details on this in the paper. To summarize, for very large datasets (in the scale of millions of data points), it is preferable to use FLQMI and GCMI. For medium scale datasets, the other SMI functions (FLVMI and LOGDETMI) can be used with the partitioning trick, which will slightly reduce the accuracy.
>
> ### Response to Other Comments:
>
> > Q1: How are confidence bars defined and computed?
>
> A1: The line plots for each strategy are computed by taking an average over three runs; hence, each point in the graph is calculated as the average of the accuracy of that strategy at the given labeled set size. We also compute the standard deviation for each point in the same manner (Ln. 286-287). The confidence bar, then, is calculated as the range spanning one standard deviation from the average.
>
> > Q2: Related, why for many plots the error bars shrink to a point (for example,figure 3(d), DIV-GCMI at labeled set size=1000), or even worse disappear (for example, figure 3(d) LOGDETMI after 1125)?
>
> A2: As one adds more data points towards the end of AL, the variance of all approaches decreases. In Figure 3(d), note that the overall accuracy is > 90%; at that point, the accuracy has pretty much converged since the results are on MNIST digit classification. Hence, there should be low variance.
>
> > Q3: Why don't we have them for image net dataset?
>
> A3:  Due to resource constraints, we ran experiments on ImageNet only once (see Ln. 287-288). However, we will add error bars for the same in the next version of the paper.
>
> > Q4: It would be more convincing if you could present experiment results with different choices of hyper parameters (for example, learning rates, batch sizes, etc.
>
> A4: Thank you for your suggestion. We will add them in the next version of the paper.
>
> ### References
>
> [a] Killamsetty, Krishnateja, et al. "GLISTER: Generalization based Data Subset Selection for Efficient and Robust Learning." Proceedings of the AAAI Conference on Artificial Intelligence. Vol. 35. No. 9. 2021.
>
> [b] Mirzasoleiman, Baharan, Jeff Bilmes, and Jure Leskovec. "Coresets for data-efficient training of machine learning models." International Conference on Machine Learning. PMLR, 2020.
>
> [c] Killamsetty, Krishnateja, et al. "GRAD-MATCH: Gradient Matching based Data Subset Selection for Efficient Deep Model Training." International Conference on Machine Learning. PMLR, 2021.
>
> [d] Gudovskiy, Denis, et al. "Deep active learning for biased datasets via fisher kernel self-supervision." Proceedings of the IEEE/CVF Conference on Computer Vision and Pattern Recognition. 2020.
>
> [e] Ash, Jordan T., et al. "Deep batch active learning by diverse, uncertain gradient lower bounds." arXiv preprint arXiv:1906.03671 (2019).

---

### Official Review · Reviewer_enP1 · 2021-07-15

**Rating:** 5
**Confidence:** 3

**Summary:**

This work attempts to address several situations that may occur during active learning processes, i.e., data imbalance, redundancy, rare classes and out-of-distribution with some submodular information measures.

**Limitations And Societal Impact:**

The authors have adequately addressed the limitations and potential negative societal impact of their work.

**Main Review:**

For methods: this work should be an application/extension of "Submodular Combinatorial Information Measures with Applicationsin Machine Learning". However, I don't agree with the application to the scenarios rare classes/data imbalance and OOD, since both rare classes/data imbalance and OOD cases be identified based on the assumption that the labels of data are known. However, in active learning processes, we assume that the labels could not be accessed at the beginning, therefore, one cannot determine if these scenarios exist. In machine learning, these exteremes are possible but in active learning, we have no idea to identify whether the unlabeled data pool containes rare classes/data imbalance, OOD scenarios.

For experiments:

1) In OOD related experiments, it's unfair to exclude OOD data points in initial labelled set, using random selection would be better.

2) In Figure. 3-5, the color choice is so poor that it’s really hard to distinguish some strategies from the others.

3) The upper bound of labeled set size is too small in some cases, e.g., in Figure. 3 (b), it does not converge like Figure. 4 (b), the upper bound of labeled set size should be increased in case that the proposed method perform good in previous stages and not as good as baseline models in latter stages.

4) The proposed SIMILAR method on standard AL (in Appendix), which is the "Achilles heel" of the proposed model, this suggests that it is difficult to put it into practical use with more general cases.


To sum up,  the submodular information measures are indeed very good ideas with strong theoretical foundations, but I'm still skeptical about applying it to active learning scenarios. Additionally, some experimental settings are not enough convincing.


/**************************/
After reading the authors' response, I decide to keep my score. Additionally, the author mentioned that "for instance, in medical imaging domains, images of cancer cells are often rare and are critical to classify correctly", but they didn't have the related experiments/case study.

**Time Spent Reviewing:**

10

---

> ### Author Response · Authors · 2021-08-10
> **Response to Reviewer enP1**
>
> Thank you very much for taking the time to review our submission and providing your valuable comments! Here are our responses to your comments
>
> > Q: However, I don't agree with the application to the scenarios rare classes/data imbalance and OOD, since both rare classes/data imbalance and OOD cases be identified based on the assumption that the labels of data are known. However, in active learning processes, we assume that the labels could not be accessed at the beginning, therefore, one cannot determine if these scenarios exist
>
> A: We agree that labels of all samples are not known beforehand. Below is a case-wise summary of how we can gauge or encounter each of the scenarios.
>
> 1. **Realizing rare classes:** i) The initial labeled set used in AL usually follows the distribution of the unlabeled set. The statistics of this set can be used to identify rare classes. ii) If the initial seed set is small, the rare classes can be realised after a few rounds of standard AL. Until these rare classes are found,  standard AL can be done using a diversity-based acquisition function like LOGDET. iii) Production-level models go through a test deployment phase. During this phase, systematically recurring errors are often found. An example is of undetected bicycles at night in an object detector (false negatives). Such recurring failure cases can be due to rare classes in the labeled set. iv) Moreover, we as users often know whether there are rare classes or if there is redundancy from domain knowledge.  For instance, in the biomedical domain, images of cancer cells are typically rarer than ones of non-cancer cells because cancer inherently is a rare disease.
>
> 2. **Realizing redundancy:** In most datasets, redundancy is naturally obvious. For example, footage from a CCTV camera and frames sampled from a self-driving car camera driving on a freeway naturally introduce redundant data. Furthermore, this can also be gauged from the initial labeled set.
>
> 3. **Realizing OOD:** Interestingly, OOD scenario is not required to be known upfront. Even if OOD data is not discovered during the labeling of the initial seed set, it can be encountered during the labeling phase of an intermediate round of active learning. The flexibility of our framework allows this conditioning to be added at any point by simply setting the private set $P \leftarrow O$ to avoid selecting OOD points similar to $O$ in the future. This is what we use in our experiments; i.e., we *do not* assume we know all the OOD points in the unlabeled set -- only a much smaller subset encountered thus far.
> A similar question was asked by reviewer yRLn. We add the answer here for completeness and will also add a discussion in the main paper.
>
> > Q1: In OOD related experiments, it's unfair to exclude OOD data points in initial labelled set
>
> A1: Though the initial labeled set does not have OOD data points, this actually does not make any noticeable difference. This is because we have an additional class called "OOD" in our model which all techniques use (Ln. 215-216). In fact, we ran an experiment where we added OOD data points in the seed set, and we saw that the SMI/SCMI functions still had similar gains over random and existing baselines.
>
> > Q2: In Figure. 3-5, the color choice is so poor that it’s really hard to distinguish some strategies from the others.
>
> A2: Thank you for pointing this out. We will update it with a better color scheme in the next version.
>
> > Q3: The upper bound of labeled set size is too small in some cases, e.g., in Figure. 3 (b), it does not converge like Figure. 4 (b)
>
> A3:  The main goal of the rare classes scenario was to improve accuracy on the rare classes. As suggested, we did run the experiment for a few more rounds until all the rare class samples were selected using the SMI functions. After that, since there were no rare-class samples left in the unlabeled set, there is no more significant gain in accuracy on the rare classes. The other baselines eventually converged with a higher labeling cost. This also proved that the SMI functions converged much earlier than the other functions in the rare classes scenario. We thank you for the suggestion and will update the plots with a larger upper bound of the labeled set size.
>
> > Q4: The proposed SIMILAR method on standard AL (in Appendix), which is the "Achilles heel" of the proposed model, this suggests that it is difficult to put it into practical use with more general cases.
>
> A4: As pointed out, SIMILAR performs *competitively* in the Standard AL setting. Also note that in the standard AL setting, SIMILAR is basically submodular function optimization -- see Ln. 122-124 (i.e., there is no external query set or private set). However, the true utility of SIMILAR is in realistic scenarios where other AL methods underperform by significant margins. As motivated in section 1 of the main paper, it is imperative to apply SIMILAR in many critical deployment scenarios which suffer due to rare classes, OOD data and redundancy. For instance, in medical imaging domains, images of cancer cells are often rare and are critical to classify correctly. As shown by our experiments, current methods are not well-equipped to do so.

---

> > ### Author Response · Authors · 2021-09-01
> > **Response to additional comments**
> >
> > Thank you for your response and additional comments. Please find our response to your comments:
> >
> > > Additionally, the author mentioned that "for instance, in medical imaging domains, images of cancer cells are often rare and are critical to classify correctly", but they didn't have the related experiments/case study.
> >
> > As a proof of concept, we evaluate our method using few representative functions on a biomedical dataset containing pre-processed pediatric chest X-ray images [a,b]. We use $\rho=20$ such that X-ray images with pneumonia are rare. The initial labeled set had $C_L + D_L=105$ and the unlabeled set had $C_U + D_U=1103$ and $R=10$. In every AL round, we added 10 data points to the labeled set. For training, we used a ResNet18 model, an SGD optimizer with an initial learning rate of 0.0003, a momentum of 0.9, and a weight decay of 5e-4. We test on the default test set provided. The results table below show the effectiveness of SMI functions:
> >
> > | Method | Initial accuracy | R1    | R2    | R3    | R4    | R5    | R6   | R7    | R8    | R9    |
> > |--------|------------------|-------|-------|-------|-------|-------|------|-------|-------|-------|
> > | FL1MI  |            53.76 |  72.9 | 73.07 |  75.3 | 73.39 |  74.9 | 80.2 |  79.6 | 82.05 | 85.09 |
> > | FL2MI  |            53.76 |  70.6 |  72.9 | 73.87 | 72.91 | 74.87 | 76.9 | 77.08 | 85.73 |  84.8 |
> > | US     |            53.76 | 68.75 |  67.5 | 67.06 | 73.87 |    75 | 75.1 |  74.3 |  74.6 |  74.7 |
> > | Random |            53.76 |  51.9 |  50.3 |  51.9 | 52.08 |  53.7 | 53.2 |  55.6 |  55.1 |    56.1 |
> >
> > In particular, notice that the SMI functions (variants of FLMI) are ~10% better compared to uncertainty sampling and more than 28% better compared to random sampling! We will add these results in the next version of the paper as a real world scenario.
> >
> > ### References
> >
> > [a] Yang, Jiancheng, Rui Shi, and Bingbing Ni. "Medmnist classification decathlon: A lightweight automl benchmark for medical image analysis." 2021 IEEE 18th International Symposium on Biomedical Imaging (ISBI). IEEE, 2021.
> >
> > [b] Kermany, Daniel S., et al. "Identifying medical diagnoses and treatable diseases by image-based deep learning." Cell 172.5 (2018): 1122-1131.

---

### Official Review · Reviewer_yRLn · 2021-07-19

**Rating:** 6
**Confidence:** 4

**Summary:**

In this paper, the authors propose a new diversity based active learning algorithm by utilizing a series of information functions in submodular optimization (SO) problems. Specifically, they consider three submodular information measures (SIM) in SO: Submodular Mutual Information (SMI), Submodular Conditional Gain (SCG) and Submodular Conditional Mutual Information (SCMI) and found that SCMI ($I_f(\mathbf{A};\mathbf{Q}|\mathbf{P})$) is the most expressive SIM such that one could tailor $\mathbf{Q}$ and $\mathbf{P}$ to represent SCG and SMI, and thus propose to use it as a “unified” acquisition function for tackling three scenarios in realistic AL: rare class due to class imbalance, redundant data and out-of-distribution(OOD) data. The basic idea is to specify $\mathbf{Q}$ and $\mathbf{P}$ depending on the desired/un-desired classes, such that one could condition on data region that they want to exclude (OOD, redundant data), while up-sample regions that one want to include (rare class) by considering the mutual information w.r.t that region. To instantiate SIM acquisition function, the authors borrow the utility function from three well known graph-based SO problems for diversity coverage: Facility location (FL), Graph cut (GC) and Log Determinant (LOGDET), and use the similarity kernels computed with pairwise cosine similarity of the last layer gradients w.r.t the model-predicted label for every pair of unlabeled data points. They experiment with synthetic datasets rooted from MNIST/CIFAR-10/ImageNet to test for the three special scenarios, and show promising results against several strong baselines including BADGE.

**Limitations And Societal Impact:**

This reviewer did not find any potential negative societal impact.

**Main Review:**

Pros
* The authors provide a systematic exploration of the application of submodular information measures from graph-based submodular optimization problems on diversity based batched AL, which is quite unique and novel.
* The proposed SCMI framework is one of the first (to this reviewer) that can be explicitly tailored for all three scenarios (rare class, OOD data, and redundancy).
* The authors conduct quite extensive benchmark experiments on various synthetic setups and show consistent out-performance on the three special scenarios in the metrics of interest.
* The paper is well structured and covers a lot of material.

Concerns
* The method seems to highly rely on the availability of example data for the desired/un-desired classes, which is possible in synthetic dataset but not always achievable in reality. For example, how do one know ahead of time which classes are the minor/rare class before having the labels available? How do one define OOD samples on the large space of images? Without human susceptions or prior knowledge, the proposed methods may face limitations when applied in real dataset.

* The rare class and OOD setup both specify a subset of samples to compute mutual information with. For rare class it uses $\mathit{R}$ which is a set of labeled rare class sample and for OOD it uses the currently labeled in-distribution data $\mathit{I}$. However, neither of the subsets is guaranteed to be covering enough sample space of the full unlabeled and relevant (non-OOD) examples. Wouldn’t this introduce bias into the AL acquisition such that it will only pick samples that are similar to the subset while leave the rest unexplored?

* There might be some concerns in the experimental setups.  If R consists of data from imbalanced classes $\mathbf{C}$ which is 10 or 20 larger than $\mathbf{D}$, how come $\mathbf{C}$ is considered as rare class? As the SIM methods oversamples data similar to R (which represents the majority classes C), it is not surprising that SIM will outperform all baselines on these “rare classes” given that the baseline models do not have access to the additional information in R. Similarly, there could be bias in the over accuracy metric which favors SIM methods.

* On the redundancy dataset, the advantage of SIM methods only starts to show up when there is large enough labeled data (as it relies on the conditioning on already-queried points to account for diversity). This means that one will have to find a sweet spot such that using the proposed method can be beneficial, which is not always possible.

* Moreover, it seems that the SIMILAR methods do not perform as well in standard AL setting (appendix figure 6), indicating potential risk of applying such biased optimization. These limitations should be addressed more carefully.
* The figures can be further improved by marking out SIMILAR-based methods as “(Ours)” as there are so many abbreviations of different methodologies and it is very easy to lose track of the result.


**Time Spent Reviewing:**

2 hours

---

> ### Author Response · Authors · 2021-08-10
> **Response to Reviewer yRLn**
>
> Thank you very much for taking the time to review our submission and providing your valuable comments! Here are our responses to your comments:
>
> > Q1: The method seems to highly rely on the availability of example data for the desired/un-desired classes, which is possible in synthetic dataset but not always achievable in reality.
>
> A1:  Interestingly, a user of this framework does not need to know all issues in the dataset beforehand and can glean example data as the AL loop progresses. Below is a case-wise summary of how we can gauge or encounter each of the scenarios.
>
> 1. **Realizing rare classes:**  i) The initial labeled set used in AL usually follows the distribution of the unlabeled set. The statistics of this set can be used to identify rare classes. ii) If the initial seed set is small, the rare classes can be realized after a few rounds of standard AL. Until these rare classes are found,  standard AL can be done using a diversity-based acquisition function like LOGDET. iii) Production-level models go through a test deployment phase. During this phase, systematically recurring errors are often found. An example is of undetected bicycles at night in an object detector (false negatives). Such recurring failure cases can be due to rare classes in the labeled set. iv) Moreover, we as users often know whether there are rare classes or if there is redundancy from domain knowledge.  For instance, in the biomedical domain, images of cancer cells are typically rarer than ones of non-cancer cells because cancer inherently is a rare disease.
>
> 2. **Realizing redundancy:** In most datasets, redundancy is naturally obvious. For example, footage from a CCTV camera and frames sampled from a self-driving car camera driving on a freeway naturally introduce redundant data. Furthermore, this can also be gauged from the initial labeled set.
>
> 3. **Realizing OOD:** Interestingly, OOD scenario is not required to be known upfront. Even if OOD data is not discovered during the labeling of the initial seed set, it can be encountered during the labeling phase of an intermediate round of active learning. The flexibility of our framework allows this conditioning to be added at any point by simply setting the private set $P \leftarrow O$ to avoid selecting OOD points similar to $O$ in the future. This is what we use in our experiments; i.e., we *do not* assume we know all the OOD points in the unlabeled set -- only a much smaller subset encountered thus far.
> We thank you for pointing this out and will add a discussion on this in the next version of the paper.
>
> > Q2:  Neither of the subsets is guaranteed to be covering enough sample space of the full unlabeled and relevant (non-OOD) examples. Wouldn’t this introduce bias into the AL acquisition?
>
> A2: That’s a great point! If the query sets in both cases remain static, then repeated AL rounds would likely exhibit the bias that you’ve described for AL selection. However, one crucial detail of our experiments that we would like to highlight is that the query set is updated with the in-distribution examples and the private set is updated with the OOD examples that are selected (Ln. 353-354). Hence, future AL selections now have a query set and a private set that are more diverse than before, which can allow better exploration of the feature space while still performing targeted selection. This is also true of the rare class experiment. As we obtain more labeled data points, the gradient embeddings (which are used in the similarity functions) become better featurizations of the data as AL progresses, and the query/ood set in the SMI/SCMI functions becomes more representative via the added data points. Both aspects ensure that we can select better points relevant to the targeted queries. Furthermore, some of the submodular (conditional) mutual information functions naturally account for diversity in the selection process. Because of this diversity, it is likely that the points selected will cover the relevant unexplored portions of the sample space.
>
> > Q3-P1:  If R consists of data from imbalanced classes  which is 10 or 20 larger than , how come  is considered as rare class?
>
> A3-P1: We recognized a typo after submission. The cardinality equations on Ln. 295 suggest that the number of rare examples is 10 to 20 times larger than the number of balanced examples. The actual experiment ensures the opposite: there are 10 to 20 times the number of balanced examples than there are rare examples (hence, the rare examples are actually “rare”). We apologize for this confusion and will fix this typo in the next version.
>
> >  Q3-P2: it is not surprising that SIM will outperform all baselines on these “rare classes” given that the baseline models do not have access to the additional information in R.
>
> A3-P2: All our baselines have access to $R$. With reference to Figure 3 in the main paper, GLISTER, FISHER and FISHER-LV are baselines that use $R$ in the same way as the SMI functions. Infact, FISHER-LV uses a $40 \times$ larger set of rare classes than our methods (Ln. 317-318), and yet it underperforms. The remaining baselines have $R$ added to their labeled set.
>
> > Q4: On the redundancy dataset, the advantage of SIM methods only starts to show up when there is large enough labeled data.
>
> A4: That is a great point! This is a consequence of the experimental setup. The initial labeled set has unique points that are disjoint from the unlabeled set. Hence, the initial rounds of AL do not have much information in the form of the conditioning (private) set to avoid redundant points. However, this issue resolves after a few rounds of active learning, when the conditioning starts to show an effect (see Figure 4c). In real-world datasets, we would expect the conditioning to show an effect earlier since there may exist data points in the labeled dataset that are semantically similar to the unlabeled dataset (which is not the case in our experiments).
>
> > Q5: Moreover, it seems that the SIMILAR methods do not perform as well in standard AL setting (appendix figure 6), indicating potential risk of applying such biased optimization.
>
> A5: As pointed out, SIMILAR performs *competitively* in the Standard AL setting. Also note that in the standard AL setting, SIMILAR is basically submodular function optimization -- see Ln. 122-124 (i.e., there is no external query set or private set). Hence, there is really no bias here in the form of a specific query set. However, the true utility of SIMILAR is in realistic scenarios where other AL methods underperform by significant margins. As motivated in section 1 of the main paper, it is imperative to apply SIMILAR in many critical deployment scenarios which suffer due to rare classes, OOD data and redundancy. For instance, in medical imaging domains, images of cancer cells are often rare and are critical to classify correctly. As shown by our experiments, current methods are not well-equipped to do so.
>
> > Q6: The figures can be further improved by marking out SIMILAR-based methods as “(Ours)”
>
> A6: Thank you for highlighting this. We will highlight our methods as you described in the next version of the paper.

---

### Decision · Program_Chairs · 2021-09-28

**Decision:**

Accept (Poster)

**Comment:**

The reviewers appreciate the paper’s general idea in a unified AL framework that addresses different AL tasks (rare class, OOD data, and redundancy), and acknowledge the technical novelty in the acquisition function which (non-trivially) generalizes existing heuristics. However, there remains concern in the presentation clarity (e.g. justification of important assumptions such as accessibility to OoD data for certain applications, the significance of contributions w.r.t. existing work) and the experimental setup. Thus the reviewers were not convinced that the method is well justified in its current state to merit acceptance for publication.

**Consistency Experiment:**

NeurIPS has a long history of experimentation. In 2014, NeurIPS ran an experiment in which 10% of submissions were reviewed by two independent committees to quantify the randomness in the review process. This year, we repeated a variant of this experiment to see how the quality of the review process has changed over time.  This paper was part of the experiment and was therefore assigned to two committees (consisting of reviewers, an Area Chair, and a Senior Area Chair) that reached independent decisions.  If both committees made the same recommendation, this recommendation was followed. If a single committee recommended acceptance, the paper was accepted (with the exception of a few cases in which the other committee identified what we considered a fatal flaw, e.g., an error in a key result).

This copy’s committee reached the following decision: **Reject**

The other committee assigned to the paper recommended **Accept (Poster)**.  You can find the other set of reviews, along with any follow up discussion with the authors here:
https://openreview.net/forum?id=x_n34KpwAvI